# Optimized design of planar solid oxide fuel cell interconnectors

**Boxiang Sun[1], Huiyu Wang[1], Songyan Zou[1], Xiang Shao [2]\***

**1** School of Mechanical Engineering, University of Leeds, Leeds, United Kingdom, **2** Nanjing Institute of Technology, School of Mechanical Engineering, Nanjing, 211167, China

\* 15289283028@163.com

**Data Availability Statement:** All relevant data are within the paper and its Supporting Information files.

**Funding:** The study was financially supported by to Postgraduate Research & Practice Innovation Program of Jiangsu Province (Grant No.

## Abstract

Solid oxide fuel cells (SOFCs) are vital for alternative energy, powering motors effi-ciently. They offer fuel versatility and waste heat recovery, making them ideal for various applications. Optimizing interconnector structures is crucial for SOFC advancement. This paper introduces a novel 2D simulation model for interconnector SOFCs, aiming to enhance their performance. We initially construct a single half-cell model for a conventional interconnector SOFC, ensuring model accuracy. Subsequently, we propose an innovative interconnector SOFC model, which outperforms the conventional counterpart in various aspects.

## 1. Introduction

### 1.1 Background and motivation

The development of human society is intricately relevant to energy exploration and innovation. From traditional fossil fuels like coal and oil to emerging sources like solar and hydrogen energy, energy plays a pivotal role in societal progress. However, overuse on fossil fuels has led to environmental issues. To ensure long-term sustainability, the global community must strive for zero-emission energy solutions.

In recent decades, the global automotive industry has seen incredible growth, with China leading in electric vehicle production and ownership [1]. While this provides convenience, it also causes energy and environmental challenges due to the heavy reliance on conventional fossil fuels [2]. Fuel cells, specifically Solid Oxide Fuel Cells (SOFCs), offer a promising solution [3]. SOFCs directly convert chemical energy into electricity and are highly efficient, making them essential in solving energy and environmental issues. They have various structures, including planar and tubular designs, each with its advantages and disadvantages. Planar SOFCs, known for their high-power density, are a leading choice in research [4]. They offer superior electrical performance and are cost-effective for large-scale production [5]. However, challenges like sealing at high temperatures and long-term stability need handling. Tubular SOFCs, with their special design, offer increased power density but require precise and complex manufacturing processes, potentially increasing costs. Nonetheless, their high-power density makes them an accessible option for high-power applications.

SJCX22_1067). The funders had no role in study design, data collection and analysis, decision to publish, or preparation of the manuscript.

The components of an SOFC, including cathode, electrolyte, anode, and metal-interconnector support [6,7], impacts performance and efficiency. Each composition has its advantages depending on specific requirements. Interconnectors play a vital role in SOFCs by distributing airflow, delivering fuel and oxygen, and outputting reaction products [8–10]. The choice of interconnector material is crucial to maintaining stability and performance [11–21].

## 1.2 Literature review

Numerical simulation is of great significance to the study of Solid oxide fuel cells (SOFCs) [22]. It provides essential tools for modeling and predicting SOFC performance, reaction mechanisms, and transport behavior [23]. Researchers have developed accurate mathematical and physical models that encompass mass transfer, heat conduction, electron/ion transport, and electrochemical reactions [24]. These models are continually refined through comparisons with experimental data to enhance accuracy.

Sun et al. [25] investigated material properties and reaction mechanisms, while macroscale simulations provide insights into overall performance and system behavior. Numerical simulations shed light on transport processes and electrochemical reactions, leading to improve fuel cell efficiency [26]. Lai et al. [27] focused on thermal management and Multiphysics field coupling effects, optimizing heat transfer paths and cooling systems to address thermal stress and thermal runaway. Optimization algorithms, proposed by Ni et al. [28], helped find design solutions to enhance SOFC performance by adjusting variables like material properties and structural parameters. In summary, numerical simulations have made significant progress in model development, scaling studies, understanding transport processes and reaction mechanisms, optimizing thermal management, and aiding in design. These results provide a solid theoretical foundation for SOFC technology development [29].The optimized design of SOFC interconnectors has become increasingly important in fuel cell research. Researchers have optimized interconnector geometry, porosity distribution, material selection, and flow channel design to improve performance and efficiency [30,31]. Conductive material studies have enhanced electrical properties [32]. Thermal efficiency and stability have been improved through thermal conduction characteristics optimization and Multiphysics field modeling [33,34].

Current research focuses on geometry, mass transfer, conductive materials, thermal management, and Multiphysics field coupling for interconnector optimization [35]. While these results provide a basis for SOFC performance enhancement, further research is needed to achieve higher power density, longer lifetime, and lower manufacturing costs [36].

## 1.3 Major contribution

The primary contribution of this work is to investigate the profound influence of interconnector design on fuel cell stack performance within solid oxide fuel cells (SOFCs). We systematically analyze and optimize various SOFC interconnector structures, providing crucial technical insights to facilitate the early adoption of planar SOFC technology.

Designing effective SOFC interconnectors is a multifaceted challenge, primarily due to their intricate flow field structures, which maintain direct control over stack mass transfer resistance. Moreover, the transformation of these designs from theory to practice is hindered by constraints imposed by laboratory conditions and fabrication technology limitations. This study figures out these complicated issues, ultimately delivering on optimized framework that enhances the overall performance and efficiency of SOFC stacks, thus promoting the development of sustainable energy conversion. Moreover, we can use a 3D printer to solve the problems associated with the difficult manufacturing of this type of interconnector.

The rest of this paper is organized as follows. Chapter 2 established a 2D simulation model for planar SOFCs using COMSOL Multiphysics, incorporating essential physical and mathematical models. The accuracy and reliability of this model were rigorously verified and validated through grid independence tests and comparisons with experimental data. In Chapter 3, we conducted a thorough analysis of mass transfer characteristics of a novel interconnector SOFC. Cloud diagram analysis was used to evaluate mass transfer performance, highlighting key differences between the novel and conventional interconnector structures. Chapter 4 focused on optimizing interconnector structural design by using the rib height as a design variable. The main aim was to enhance the transfer performance and efficiency of interconnectors. An optimization model was developed using COMSOL Multiphysics.

## 2. Materials and methods

### 2.1. Planar SOFC two-dimensional model construction

This chapter will focus on the two-dimensional modeling of a SOFC, from an explanation of the basic structure of the SOFC to a detailed description of the mathematical modeling equations of the SOFC, which describe the physical and chemical processes that take place in the SOFC. Through numerical simulation and solving these equations, key information about the performance and behavior of SOFCs is obtained to further understand and optimize the design and operation of SOFCs. The anode, electrolyte and cathode in this SOFC form a PEN (Positive Electrode–Electrolyte–Negative Electrode) assembly [37]. The conventional interconnector design connects the anode surface of one cell to the next cathode surface. In this structure, the anode is designed with channels for fuel gas, while the cathode has channels for air flow.

### 2.2 Physical modeling

In order to better study the flow and mass transfer characteristics between the porous anode and the gas channel in a conventional interconnector SOFC, a two-dimensional SOFC model is proposed in this paper, and the following assumptions are made to simplify the model: a) The whole fuel cell system is isothermal, and radiative heat transfer and energy transfer effects are not considered. b) The gases inside the fuel channels and porous electrodes are considered as ideal gases (this assumption is valid in the SOFC operating temperature range of 973~1473 K). c) The electrochemical reaction of hydrogen occurs instantaneously at the electrode/electrolyte interface. d) The operating pressure of the fuel cell is maintained at 1 atm. e) A conventional gas channel is used for the analysis, and the flow state of the fuel gas in the channel is laminar. f) The fuel gas is a mixture of hydrogen, water vapor and argon, where argon is mainly used to regulate the molar fraction of hydrogen. g) The operating voltage of the fuel cell is kept constant, and the current in the anode is evenly distributed. h) The fuel cell is in a steady state operation process.

The model uses a fuel gas mixture of hydrogen (H2), water vapor (H2O), and argon (Ar) to study flow and mass transfer under varying hydrogen molar fractions. Fig 1 shows the simplified interconnector, while for the novel interconnector SOFC (Fig 2), the interconnector features gas channels and crossbars ('ribs') vertical to gas flow, causing significant disturbance.

In this paper, the method of drawing the geometry directly in COMSOL software is used for drawing, and the drawn geometry is the SOFC 2D model, and the geometric parameters of the model are shown in Table 1.

In this study, both the Navier-Stokes (N-S) equations and the component transport equations are solved in order to further understand the flow characteristics and mass transfer

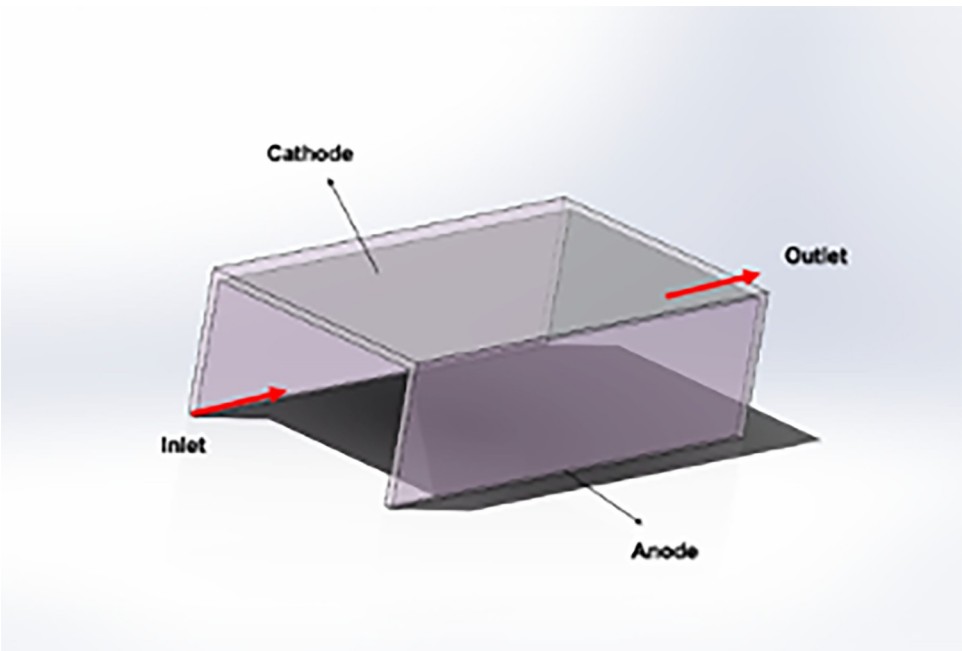

**Fig 1. Schematic diagram of a conventional interconnector unit.**

properties of the anode of a planar solid oxide fuel cell (SOFC). The governing equations for each physical equation in the two computational regions of the fuel gas channel and the porous anode, as well as their boundary conditions, are detailed below.

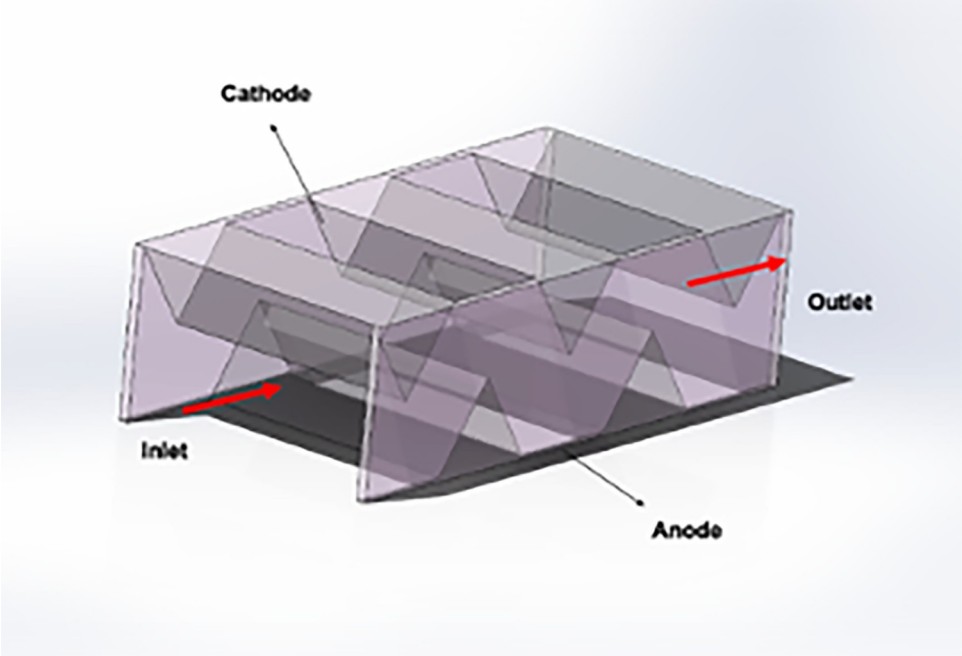

**Fig 2. Schematic diagram of the novel interconnector unit.**

**Table 1. Model geometric parameters.**

| Typology | Depth of porous anode $d_1$/m | Length of Porous anode $l_1$/m | Channel length $l_2$/m | Channel depth $d_2$/m | Rib height $h_1$/m | Rib base width $d_3$/m |
|---|---|---|---|---|---|---|
| Conventional SOFC | $0.6e^{-3}$ | $2e^{-2}$ | $2e^{-2}$ | $2e^{-3}$ | - | - |
| Novel SOFC | $0.6e^{-3}$ | $2e^{-2}$ | $2e^{-2}$ | $2e^{-3}$ | $1e^{-4}$ | $2e^{-3}$ |

## 2.3. The continuity equation and its boundary conditions

**2.3.1. Continuity equations.** In this study, we consider the continuity equation to describe the nature of mass conservation in flow processes. The continuity equation is available to both ideal and real fluids and is used in the steady-state governing equations for fuel gas channels and porous anode regions. The steady-state continuity equation can be expressed as:

In the fuel gas passage area:

$$\nabla \cdot (\rho u) = 0, \tag{2.1}$$

In the porous anode area:

$$\nabla \cdot (\varepsilon \rho u) = 0. \tag{2.2}$$

where $\rho$ is the density of the fluid, $u$ is the velocity vector of the fluid, $\varepsilon$ is the effective porosity of the porous medium, and $\nabla$ denotes the dispersion operator.

In the case of SOFCs, which are usually operated at higher temperatures (set to 1023 K in this paper), the gas mixture in the cell can be considered as an ideal gas. In the two calculation regions, the fuel gas channel and the porous anode, the fuel consists of $H_2$, $H_2O$ and Ar, and the chemical reactions and the generation of new substances are not considered in the model. Therefore, the source term S can be set to 0 in the continuity equation because the gas mass remains conserved during the flow.

**2.3.2. Boundary condition.** At the interface between the fuel gas channel and the porous anode, a continuity boundary condition was chosen for this study. This means that at the interface, the mass flow rate and mass fraction need to be continuous. Specifically, at the interface, the flow rates and component mole fractions in the fuel gas channel and porous anode may satisfy the continuity condition. For the other boundaries, this study sets the boundary condition of no slip at the walls. This means that at these boundaries, the velocity of the fluid is equal to zero (u = 0), i.e., the fluid is relatively stationary with respect to the wall and there is no slip. This boundary condition assumes a strong viscous effect in the vicinity of the wall, resulting in the fluid adhering to the wall with no sliding.

## 2.4. Momentum equations and its boundary conditions

**2.4.1. Momentum equations and boundary conditions in fuel gas channels.** (1) Governing equation (math.)

In this paper, considering that the gas flow rate in the SOFC fuel gas channel is low and the corresponding Reynolds number is small, the flow state of the fuel gas in the channel can be regarded as laminar flow. In order to depict the momentum conservation in the fuel channel, the Navier-Stokes equations [38] are used in this paper:

$$\nabla \cdot j_i + \rho (u \cdot \nabla) \omega_i = R_i. \tag{2.3}$$

where $\nabla \cdot j_i$ is the divergence of the flux $j_i$ for species $i$, $\rho$ is the density of the medium, ($u \cdot$

$\nabla)\boldsymbol{\omega}_i$ is the convective transport term for the species $i$, and $\boldsymbol{R_i}$ is the net production rate of species $i$ due to chemical reactions.

(2) Boundary condition

The fuel channel inlet takes the velocity boundary condition [39]:

$$\boldsymbol{u} = \boldsymbol{U_{in}} \cdot \boldsymbol{n}. \tag{2.4}$$

where $\boldsymbol{u}$ is the velocity vector, $\boldsymbol{U_{in}}$ is the inlet velocity magnitude, and $\boldsymbol{n}$ is the unit normal vector pointing outward from the inlet surface.

The fuel channel outlet takes the pressure boundary condition [40]:

$$\boldsymbol{\mu}(\nabla\boldsymbol{u} + (\nabla\boldsymbol{u})^T)\boldsymbol{n} = \boldsymbol{0}, \tag{2.5}$$

$$\boldsymbol{p} = \boldsymbol{p_{out}}. \tag{2.6}$$

where $\boldsymbol{\mu}$ is the dynamic viscosity of the fluid, $\nabla\boldsymbol{u}$ is the velocity gradient tensor, $(\nabla\boldsymbol{u})^T$ is the transpose of the velocity gradient tensor, and $\boldsymbol{n}$ is the unit normal vector to the surface. Also, $\boldsymbol{p}$ represents the pressure within the fluid, and $\boldsymbol{p_{out}}$ is the specified outlet pressure.

For the interface between the fuel gas channel and the porous anode, we used a continuity boundary condition; for the other boundaries, we chose a boundary condition with no slip at the wall [41]

$$\boldsymbol{u} = \boldsymbol{0} \tag{2.7}$$

where $\boldsymbol{u}$ is the velocity vector.

**2.4.2 Momentum equations and boundary conditions for porous anode region.** (1) Governing equation (math.)

In this paper, Brinkman's equation [42] is used to solve the flow field in the porous anode region with the following expression:

$$\boldsymbol{\rho}(\mathbf{u} \cdot \nabla)\mathbf{u} = \nabla \cdot [-\boldsymbol{\rho}\mathbf{l} + \mathbf{K}] + \mathbf{F}. \tag{2.8}$$

where $\boldsymbol{\rho}$ is the density of the fluid, $\mathbf{u}$ is the velocity vector of the fluid, $\nabla \cdot$ is the divergence operator, $\mathbf{K}$ is the viscous stress tensor, and $\mathbf{F}$ is the body force per unit volume acting on the fluid.

(2) Boundary condition

For the interface between the fuel gas channel and the porous anode, we used a continuity boundary condition. For the other boundaries, we set the boundary condition of no slip on the walls [41]:

$$\boldsymbol{u} = \boldsymbol{0}. \tag{2.9}$$

where $\boldsymbol{u}$ is the velocity vector.

**2.4.3. Mass transfer equations and their boundary conditions.** (1) Governing equation (math.)

In this work, considering the multicomponent nature of SOFC fuel-side gas and air-side gas, the mass transfer process between the components might be considered. The mass transfer is influenced by two main factors: the mutual diffusion between the reacting gas components (including molecular and convective diffusion), and the influence of the porous medium on the individual reacting gases (including Knudsen diffusion). In this paper, the equations set in

COMSOL for mass transfer and free and porous media flow are used as follows [43]:

$$\nabla \cdot j_i + \rho(\boldsymbol{u} \cdot \nabla)\omega_i = R_i, \qquad (2.10)$$

$$j_i = -\left( \rho D_i^f \nabla \omega_i + \rho \omega_i D_i^f \frac{\nabla M_n}{M_n} - j_{c,i} + D_i^T \frac{\nabla T}{T} \right), \qquad (2.11)$$

$$M_n = \left( \sum_i \frac{\omega_i}{M_i} \right)^{-1}, j_{c,j} = \rho \omega_i \sum_k \frac{M_i}{M_n} D_k^f \nabla x_k. \qquad (2.12)$$

where $D_i^f$ is the mass diffusivity of species $\boldsymbol{i}$, $\omega_i$ is the mass fraction of species $\boldsymbol{i}$, $M_n$ is the molar mass of the neutral component, $D_i^T$ is the thermal diffusion coefficient, and $T$ is the temperature. $j_{c,j}$ is the molar flux of species $\boldsymbol{i}$, $D_k^f$ is the binary diffusivity between species $k$ and the neutral component.

(2) Boundary condition

For the interface between the fuel gas channel and the porous anode, a continuity boundary condition was used. For the other boundaries, the boundary conditions were set as no-slip boundary conditions on the walls [41]:

$$u = \boldsymbol{0} \qquad (2.13)$$

where $\boldsymbol{u}$ is the velocity vector.

**2.4.4. Numerical methods.**   (1) Define material properties

In the 2D simulation of a conventional SOFC interconnector, defining material properties for each component (anode, cathode, electrolyte, etc.) is crucial. These properties include electrical conductivity, thermal conductivity, and the coefficient of thermal expansion, all of which significantly influence the model's behavior and performance. The appropriate expansion model and coefficients are selected based on material-specific parameters like porosity, permeability, dynamic viscosity, and diffusion coefficients, see Table 2.

The two-dimensional numerical modeling gives an understanding of gas velocity and pressure distribution within the SOFC. By comparing the flow and mass transfer characteristics of the conventional and novel interconnector models under similar or different conditions, we assess the improvement achieved by the novel interconnector design.

(2) Create a grid:

Creating a grid is crucial for simulating a conventional interconnector SOFC. First, choose an appropriate mesh type (structured or unstructured) based on software capabilities and geometry complexity. Next, determine mesh density, with finer meshing in key areas like electrode and electrolyte interfaces for precision. Finally, evaluate mesh quality, monitoring metrics like distortion and aspect ratio. This paper utilizes a free triangle mesh type, as shown in Table 3.

(3) Set the boundary conditions:

Setting up a simulation model requires defining appropriate boundary conditions and physical fields. Various boundary conditions like fixed values, transfers, and responses can be

**Table 2. Conventional SOFC constant and scalar expression settings.**

| Physical quantity | Argument | Unit |
|---|---|---|
| Porosity | 0.5 | |
| Permeability | $1e^{-12}$ | $m^2$ |
| Dynamic viscosity | $1e^{-3}$ | Pa*S |
| Diffusion coefficient | $1e^{-5}$ | $m^2/S$ |

**Table 3. Grid cell size parameters.**

| Typology | Unit size (max/min) | Maximum unit growth rate | curvature factor | narrow-area resolution |
|---|---|---|---|---|
| Conventional SOFC | $8e^{-5}/4e^{-6}$ | 1.15 | 0.3 | 1 |
| Novel SOFC | $5.6e^{-5}/8e^{-7}$ | 1.1 | 0.25 | 1 |

chosen depending on the simulation's purpose. Fixed value conditions maintain constant parameters, while boundary transfer conditions allow interactions with neighboring regions. Boundary response conditions dynamically adjust based on the model's internal changes. Additionally, setting initial conditions for temperature and concentration is crucial. These initial conditions should closely resemble real operating conditions to ensure accurate simulation results. The specific parameter settings and experimental conditions are depicted in Table 4.

(4) Solve and simulate:

The model is solved and simulated using COMSOL Multiphysics, with different solution algorithms available, such as finite element, finite difference, or finite volume methods. The choice depends on complexity, efficiency, and accuracy needs. The model is transformed into numerical form according to its equations and conditions. During simulation, monitored parameters are recorded, which can be performed to enhance system behavior and performance. Simulation results are rigorously analyzed and provide other guidance for further research and applications. It is important to select appropriate solution algorithms and methods, monitor relevant quantities, and interpret results for a comprehensive understanding of system behavior and performance.

## 2.5. Verification of grid independence

This study employs grid refinement for regions with significant physical quantity gradients. Four grid sets were validated using the average flow velocity at gas channel and porous anode interfaces as the primary metric. As the grid number increases, flow velocity profiles gradually stabilize, with minimal errors observed at 12950 and 16050 grid cells. To balance result accuracy, computational cost, and duration, 12950 grid cells were chosen, as confirmed by grid independence verification in Fig 3 and Table 5.

## 2.6. Model validation

To validate the model's accuracy and reliability, this model is compared with the straight-through channel SOFC model used by Yakabe et al. [44]. Also, the 2D numerical simulation results of the model are compared with experimental data from Yakabe et al. [44] and the 2D numerical model by Tseronis et al. [45]. Fig 4 illustrates the comparison, focusing on hydrogen mole fraction ($H_2$) at the gas channel inlet, where argon (Ar) is used to control $H_2$

**Table 4. Model run parameters.**

| Parameter/unit | Numerical value |
|---|---|
| Operating temperature /K | 1023 |
| Operating pressure /Pa | 101325 |
| Inlet velocity /m·s$^{-1}$ | 2 |
| Given current density /A·m$^{-2}$ | 3000 |
| hydrogen, $c_{H_2}^{in}$/mol · L$^{-1}$ | 11.631 |
| water, $c_{H_2O}^{in}$/mol · L$^{-1}$ | 0.24231 |
| argon, $c_{Ar}^{in}$/mol · L$^{-1}$ | 0.036347 |

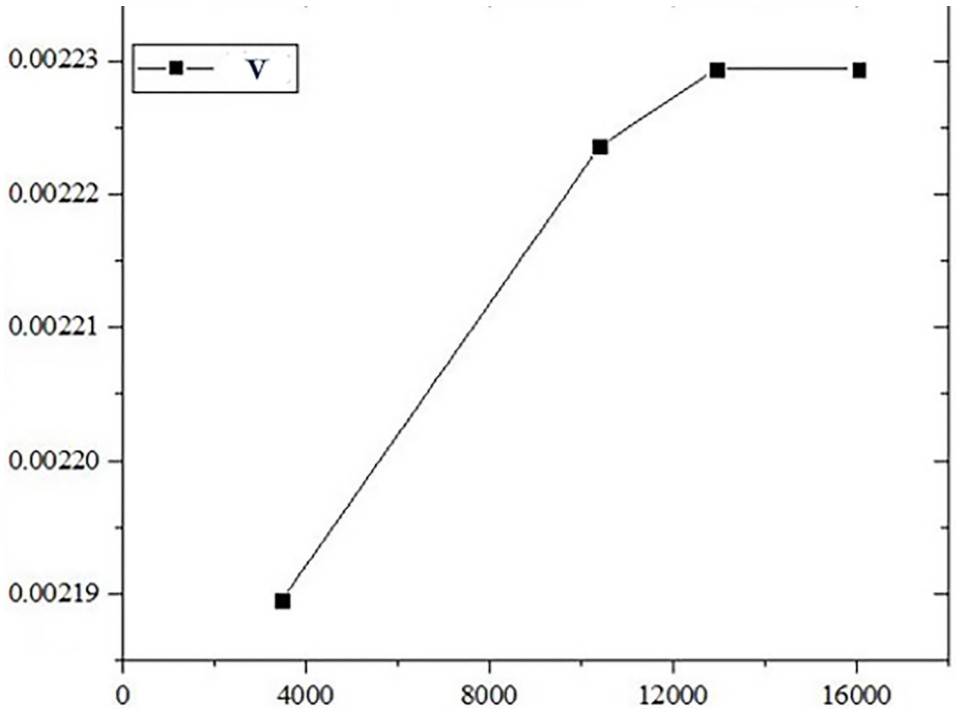

**Fig 3. Verification of grid independence.**

concentration. The calibrated model in this paper exhibits a maximum deviation of 11.7% from Yababe et al.'s experimental results [44] and 1.63% from Tseronis et al.'s numerical simulation [45]. These discrepancies can be attributed to the two-dimensional simplification, potential graphical representation errors in literature data extraction, and minor meshing variations. Despite these factors, it is evident that the model in this paper provides improved predictions of SOFC flow and mass transfer characteristics with different interconnectors.

## 2.7. Study of flow and mass transfer characteristics of SOFC with novel interconnectors

**2.7.1. Model parameters.** This section aims to compare the impact of conventional and novel interconnectors on flow and mass transfer characteristics within a planar SOFC. A two-dimensional model is employed, featuring both the conventional and novel interconnector planar SOFC models. The novel interconnector model introduces innovations, incorporating ribs spaced at specific intervals to enhance performance. Fig 5 illustrates the novel SOFC interconnector model, and Table 6 outlines its dimensional and physical parameters.

**2.7.2. Structural design of three new types of interconnectors.** In this paper, we designed three interconnectors with multiple structural parameters to investigate the influence of rib height on SOFC characteristics through numerical simulations. We compared the cloud

**Table 5. Verification of grid independence.**

| Meshes | 3468 | 10412 | 12950 | 16050 |
|---|---|---|---|---|
| Number of degrees of freedom | 9311 | 27136 | 33611 | 41511 |
| u/m s$^{-1}$ | 0.0021895 | 0.0022236 | 0.00294 | 0.0022294 |

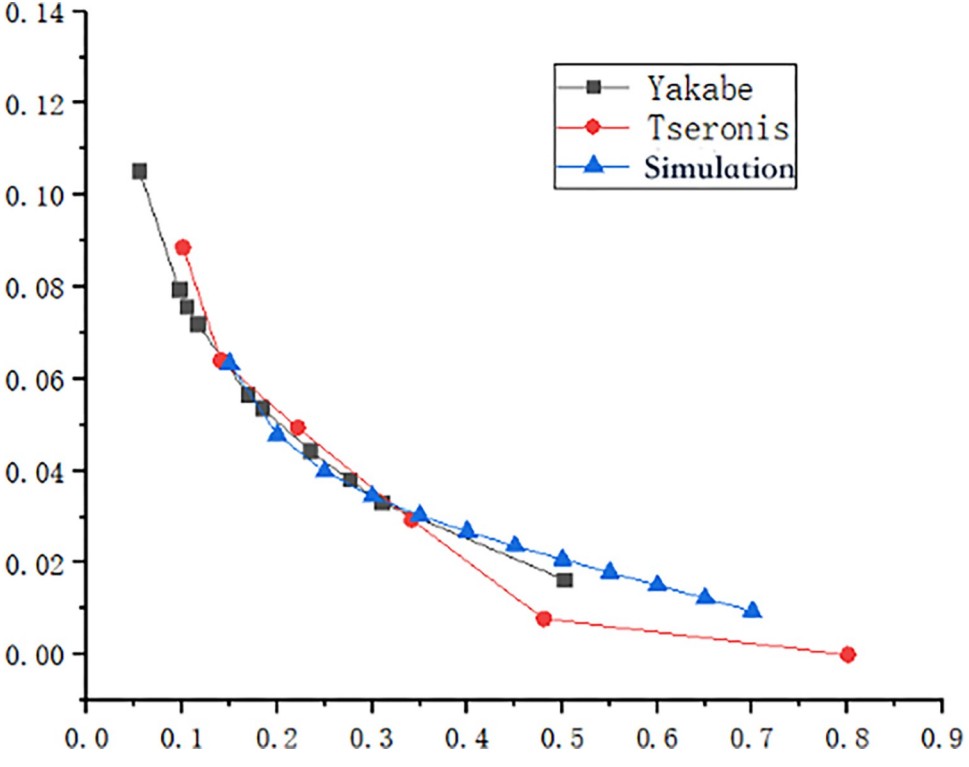

**Fig 4. Model calibration.**

diagrams of these three models and conducted a quantitative analysis of mass transfer characteristics. The primary distinction among these interconnectors lies in the height of the ribs, which increases progressively from type I to type III. Detailed parameters for the three models are provided In Table 7, and the schematic diagrams of their structures are illustrated in Figs 6–8.

## 3. Results

### 3.1. Flow characteristics

**3.1.1. Velocity field.** The velocity field analysis of the conventional SOFC interconnector model is shown in Fig 9. It demonstrates the flow patterns and speed distribution within the fuel gas channel. The novel SOFC interconnector model, depicted in Fig 10, reveals significant differences due to the rib structure, which enhances turbulence and improves mass transfer.

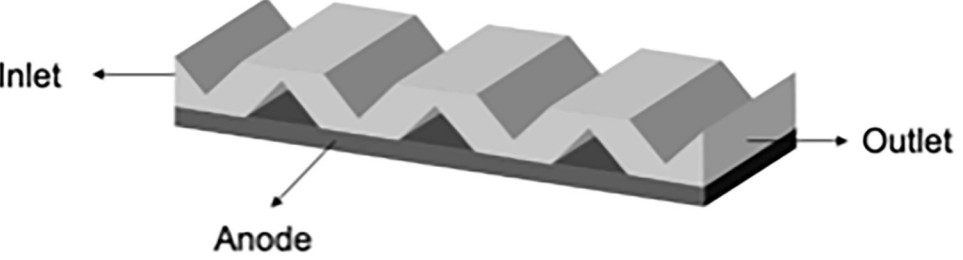

**Fig 5. Geometric model of the novel interconnector.**

**Table 6. Model dimensional and physical properties parameters.**

| Parameter/Unit | Numerical value |
|---|---|
| Model length /m | $2e^{-2}$ |
| Fuel channel height /m | $2e^{-3}$ |
| Anode thickness /m | $0.6e^{-3}$ |
| Anode porosity /% | 0.5 |
| Anode aperture /m | $1e^{-12}$ |
| Triangle bottom length /m | 0.002 |
| High degree /m | $1e^{-4}$ |

**Table 7. Model parameters.**

| Type | Rib height /m | Rib base width /m |
|---|---|---|
| I | $5e^{-5}$ | $2e^{-3}$ |
| II | $1e^{-4}$ | $2e^{-3}$ |
| III | $1.2e^{-3}$ | $2e^{-3}$ |

**3.1.2. Pressure field.** The pressure distribution within the conventional and novel SOFC interconnector models is illustrated in Figs 11 and 12, respectively. The novel interconnector shows a higher pressure gradient, which can enhance the driving force for gas flow and improve the overall performance of the fuel cell.

**3.1.3. Mass transfer characteristic.** Mass transfer characteristics are critical for the efficiency of SOFCs. Figs 13–16 illustrate the distribution of molar concentrations of hydrogen and water in both conventional and novel SOFC interconnector models. The novel design enhances the uniformity and rate of hydrogen distribution, leading to improved electrochemical reactions and overall performance.

## 3.2. Quantitative analysis of mass transfer characteristics

This section quantitatively analyzes hydrogen molar fraction in both 2D conventional and novel interconnector SOFC models. It aims to assess the impact of the novel interconnector by comparing differences in hydrogen molar fraction between the two models. The aim is to understand how the novel interconnector affects hydrogen transport efficiency and related

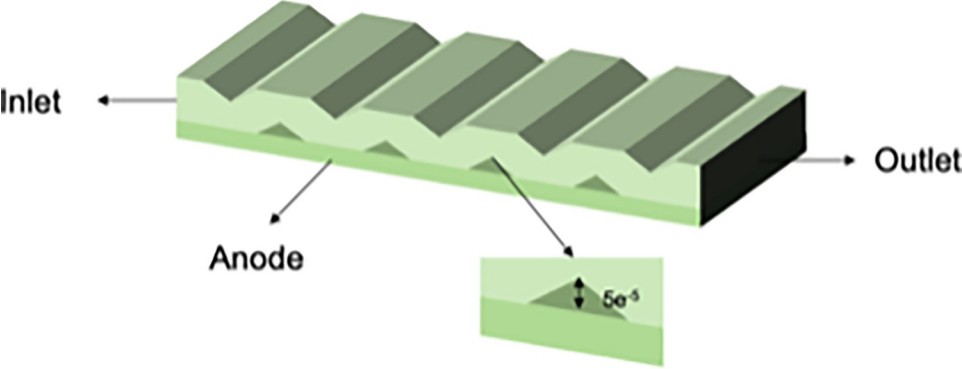

**Fig 6. Three-dimensional model of SOFC interconnector type I.**

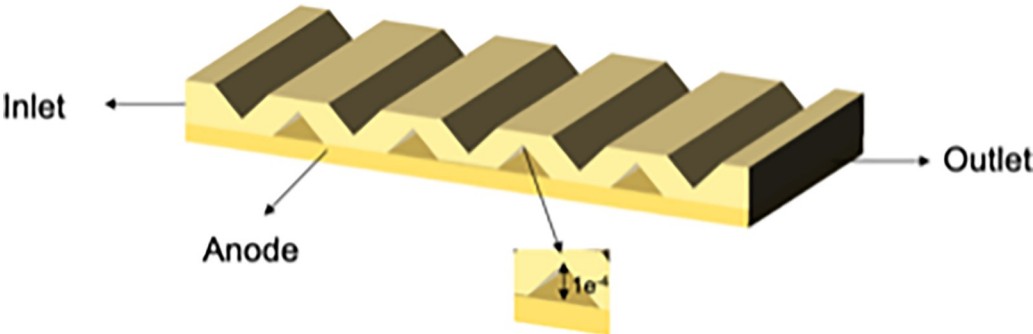

**Fig 7. Three-dimensional model of SOFC interconnector type II.**

performance optimizations. These results are crucial for optimizing SOFC interconnector design and developing fuel cell technology in sustainable energy applications.

**3.2.1. Porous anode hydrogen mole fraction distribution.** The hydrogen molar fraction within the porous anode is shown in Figs 17 and 18 for conventional and novel interconnector models, respectively. The novel design improves hydrogen distribution, reducing concentration gradients.

**3.2.2. Hydrogen molar fraction distribution in the flow field channel.** Figs 19 and 20 illustrate the hydrogen molar fraction distribution in the flow field channels of the conventional and novel SOFC interconnectors. The novel interconnector shows a more uniform and higher hydrogen concentration, promoting efficient fuel utilization.

**3.2.3. Hydrogen molar fraction distribution in the intermediate section.** The hydrogen molar fraction distribution in the intermediate section is critical for understanding the efficiency of hydrogen transport within the SOFC interconnector. Figs 21 and 22 illustrate the hydrogen molar fraction distribution for the conventional and novel SOFC interconnector models, respectively.

**3.2.4. Mole fraction distribution of hydrogen in the entrance/exit cross-section.** Analyzing the mole fraction distribution of hydrogen at the entrance and exit cross-sections of the SOFC interconnector is vital for understanding the initial and final hydrogen concentrations, which influence the overall efficiency and performance of the fuel cell. Figs 23–26 illustrate these distributions for both the conventional and novel SOFC interconnector models.

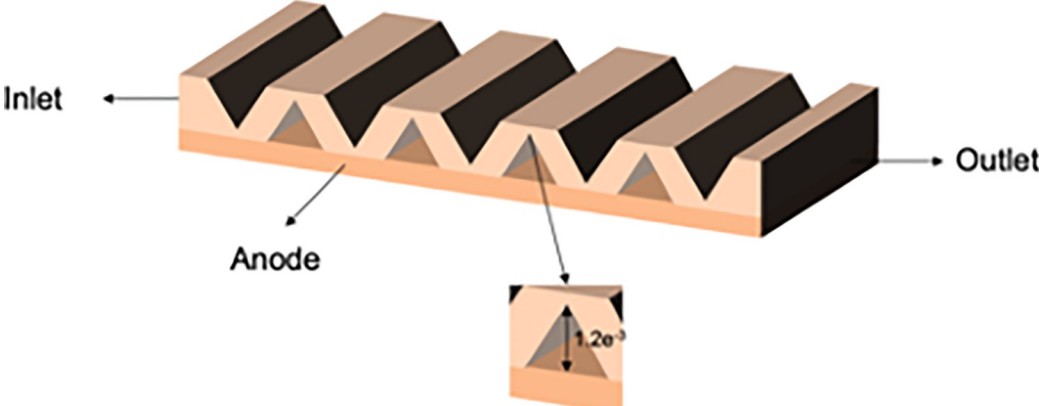

**Fig 8. Three-dimensional model of SOFC interconnector type III.**

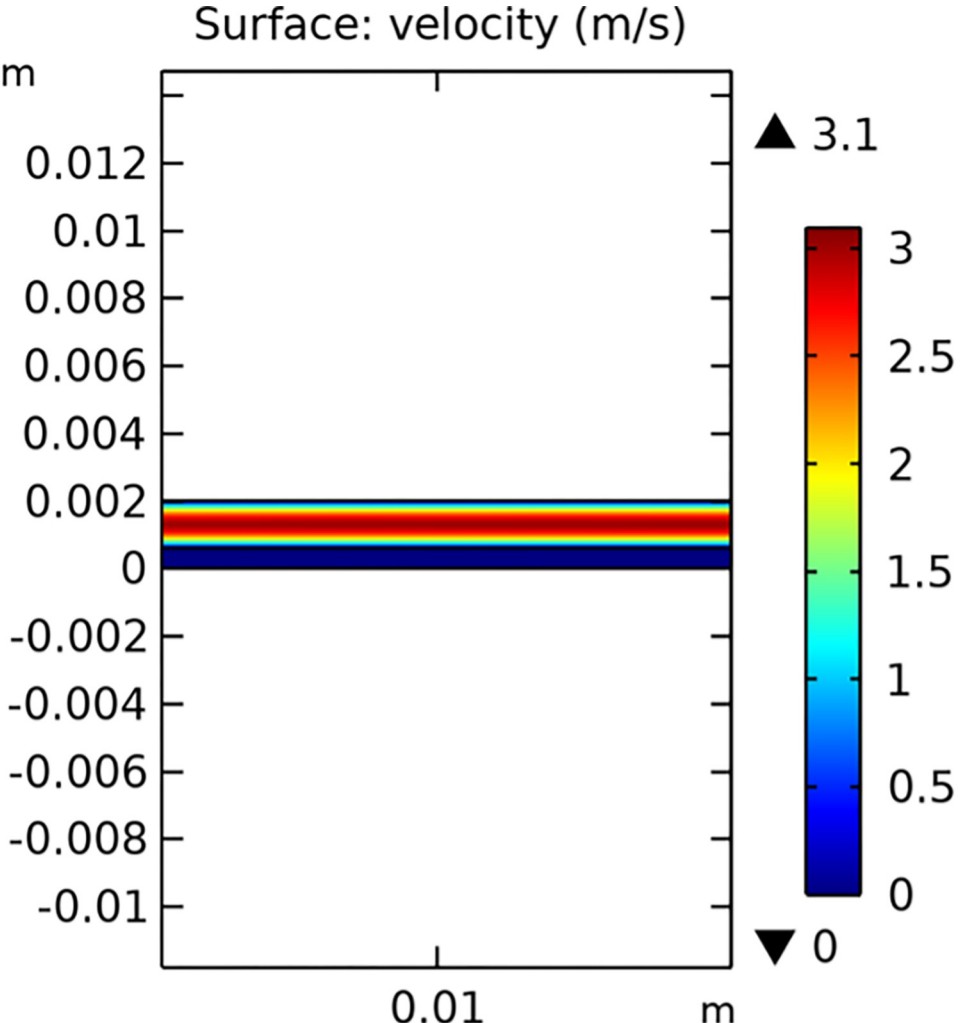

**Fig 9. Velocity distribution in conventional SOFC interconnector model (in m/s).**

**3.2.5. Porous anode interface hydrogen mole fraction distribution.** The distribution of hydrogen mole fraction at the porous anode interface is critical for understanding the interaction between the fuel gas and the porous anode, which directly influences the electrochemical performance of the SOFC. Figs 27 and 28 show the hydrogen mole fraction distribution at the interface for the conventional and novel SOFC interconnector models, respectively.

## 3.3. Flow characteristics in three types of novel interconnector

**3.3.1. Velocity field.** The velocity field within the interconnector is a key parameter that influences the mass transfer and electrochemical reactions in the SOFC. Figs 29–31 illustrate the velocity distribution for three different types of novel interconnectors.

**3.3.2. Pressure field.** The pressure field within the interconnector is a crucial parameter that affects the overall performance of the SOFC. Figs 32–34 present the pressure distribution for the three types of novel interconnectors, highlighting the differences in pressure profiles and their implications on the flow dynamics.

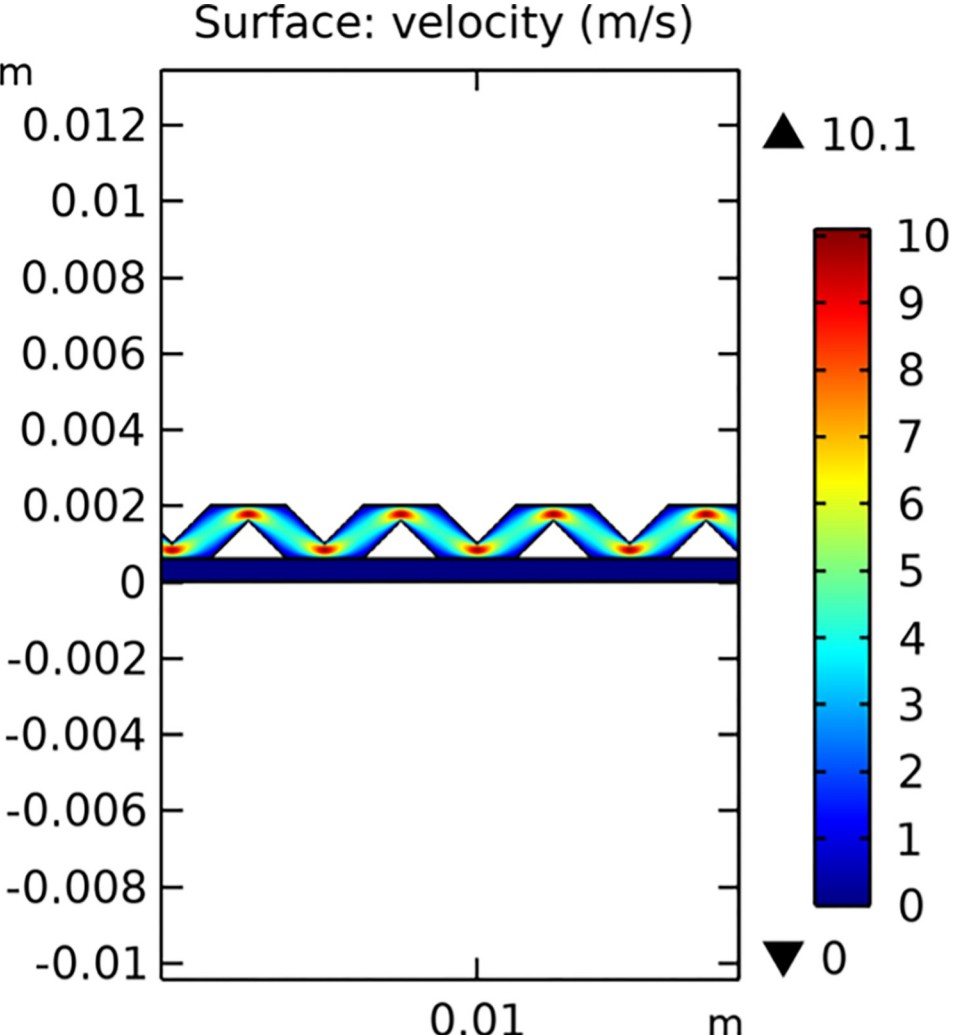

**Fig 10. Velocity distribution in novel SOFC interconnector model (in m/s).**

**3.3.3 Mass transfer characteristics.** Mass transfer characteristics are essential for the efficient operation of SOFCs as they directly influence the rate of electrochemical reactions and overall performance. Figs 35–37 illustrate the hydrogen concentration distribution for the three types of novel interconnectors.

## 3.4. Quantitative analysis of mass transfer characteristics

**3.4.1. Porous anode hydrogen mole fraction distribution.** The distribution of hydrogen mole fraction within the porous anode is critical for the performance and efficiency of the SOFC. Figs 38–40 illustrate the hydrogen mole fraction distribution for three types of novel interconnectors within the porous anode.

**3.4.2. Hydrogen mole fraction distribution in fluid channels.** The distribution of hydrogen mole fraction within the fluid channels is essential for evaluating the efficiency of hydrogen transport and utilization in the SOFC. Figs 41–43 illustrate the hydrogen mole fraction distribution for the three types of novel interconnectors within the fluid channels.

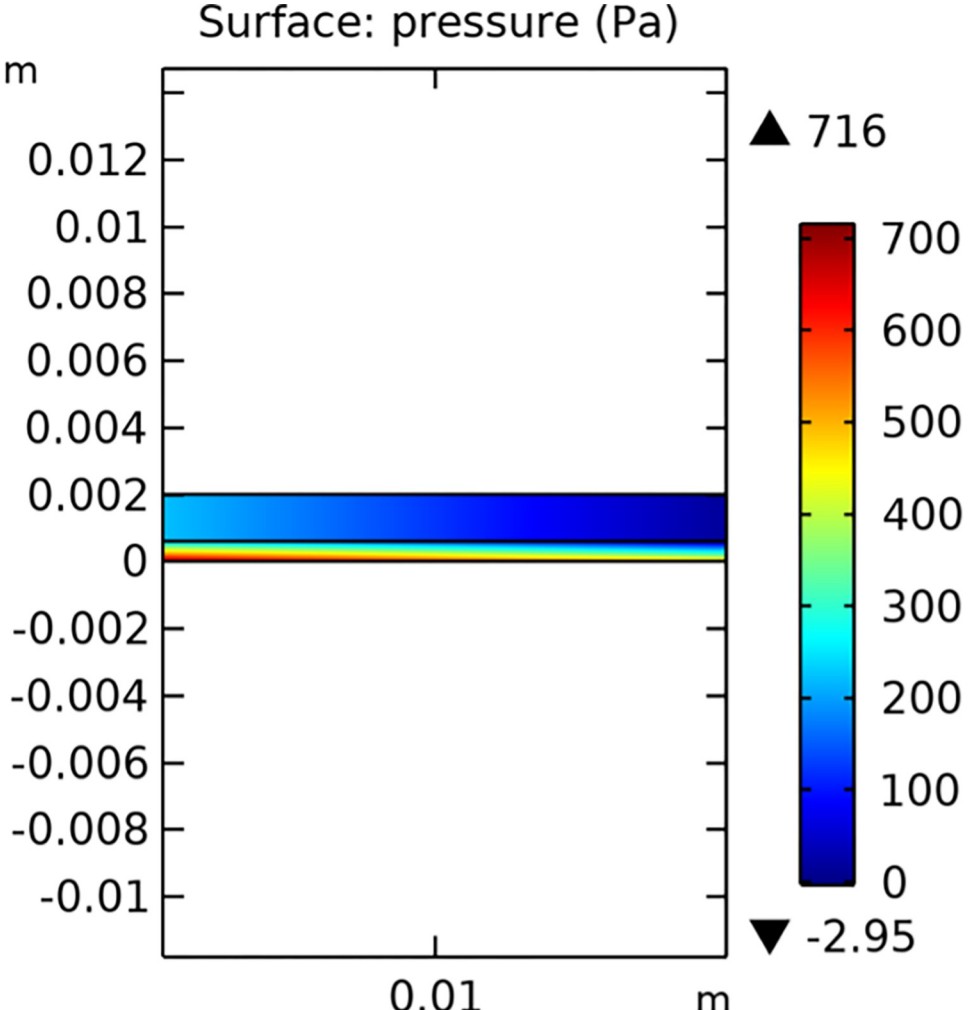

**Fig 11. Pressure distribution in conventional SOFC interconnector model (in Pa).**

**3.4.3. Mole fraction distribution of hydrogen in the inlet cross-section.** The distribution of hydrogen mole fraction at the inlet cross-section is critical for understanding the initial hydrogen supply conditions and their impact on SOFC performance. Figs 44–46 illustrate the hydrogen mole fraction distribution at the inlet cross-section for the three types of novel interconnectors.

## 4. Discussion

The novel interconnector significantly altered the flow dynamics in the fuel gas channel compared to the conventional interconnector. It induced vortex generation and increased boundary layer perturbation, resulting in a notable rise in maximum flow velocity. Specifically, the maximum flow velocity in the gas channel increased from 3.1 m/s (conventional) to 10.1 m/s (novel), marking 3.27 times increase. Additionally, the maximum flow rate through the porous electrode rose from 2.98 $e^{-3}$ m/s (conventional) to 0.013 m/s (novel), 4.36 times increase. These results demonstrate the novel interconnector's effectiveness in enhancing flow velocity and mass transfer rates within SOFC's fuel gas channel and porous electrode.

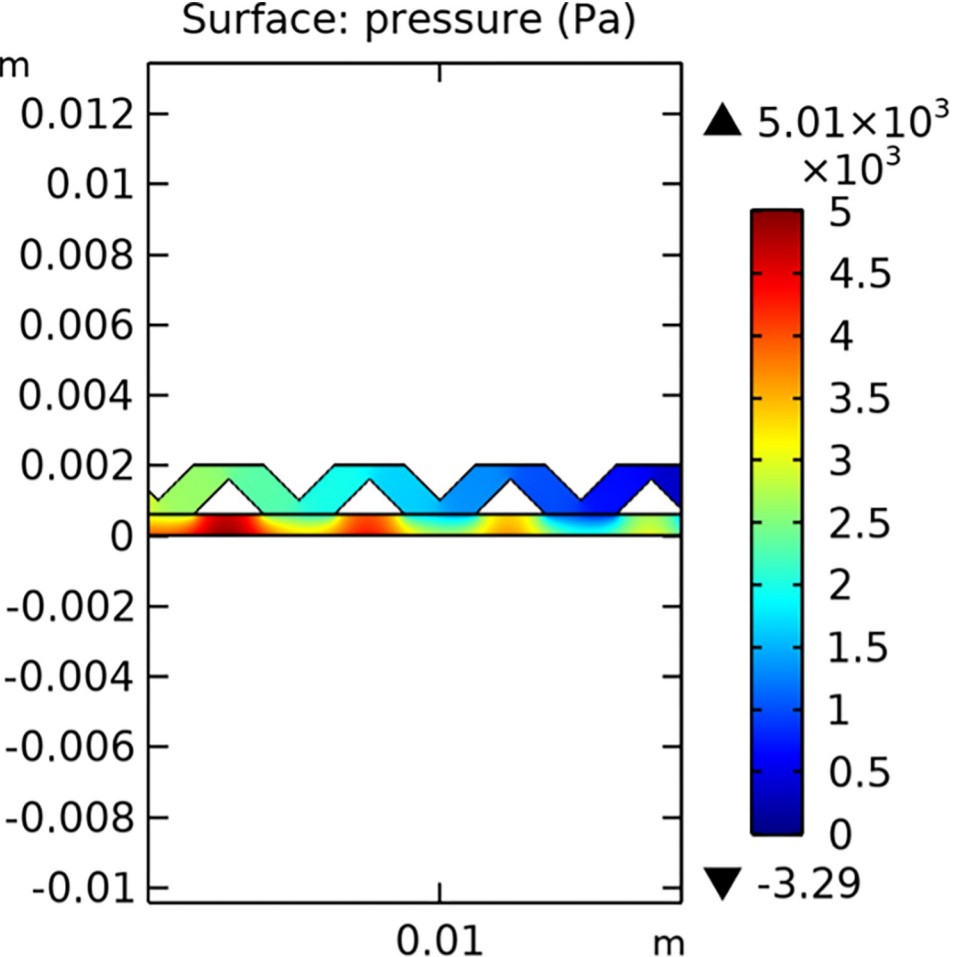

**Fig 12. Pressure distribution in novel SOFC interconnector model (in Pa).**

The novel SOFC interconnector model, featuring ribbed structures, exhibits a notable increase in pressure loss within the gas channel compared to the conventional model. Isobar diagrams reveal concentrated pressure gradients, especially below the lower rib, indicating a localized pressure difference. Consequently, it is suggested that the porous anode region below the lower rib experiences a faster flow rate. This suggests that the presence of ribs in the novel interconnector model leads to uneven pressure distribution in the gas channel and increased flow rates in specific porous anode areas. The novel interconnector improves flow rates in both the fuel gas channel and porous anode, enhancing convective diffusion within the porous anode, with negligible pressure drop in practical applications.

Comparing Figs 17 and 18, we observed a decrease in hydrogen molar fraction within the porous anodes of both the conventional and novel SOFC interconnectors. However, this decrease was more pronounced in the novel SOFC interconnector. In addition, the novel interconnector's ribbed design introduced some interference with the hydrogen molar fraction, ultimately resulting in an increased average hydrogen molar fraction. These findings highlight the novel SOFC interconnector's advantage in achieving a more homogeneous hydrogen distribution within the porous anode, with the ribs contributing to an overall increase in average hydrogen molar fraction. In comparing the hydrogen molar fraction

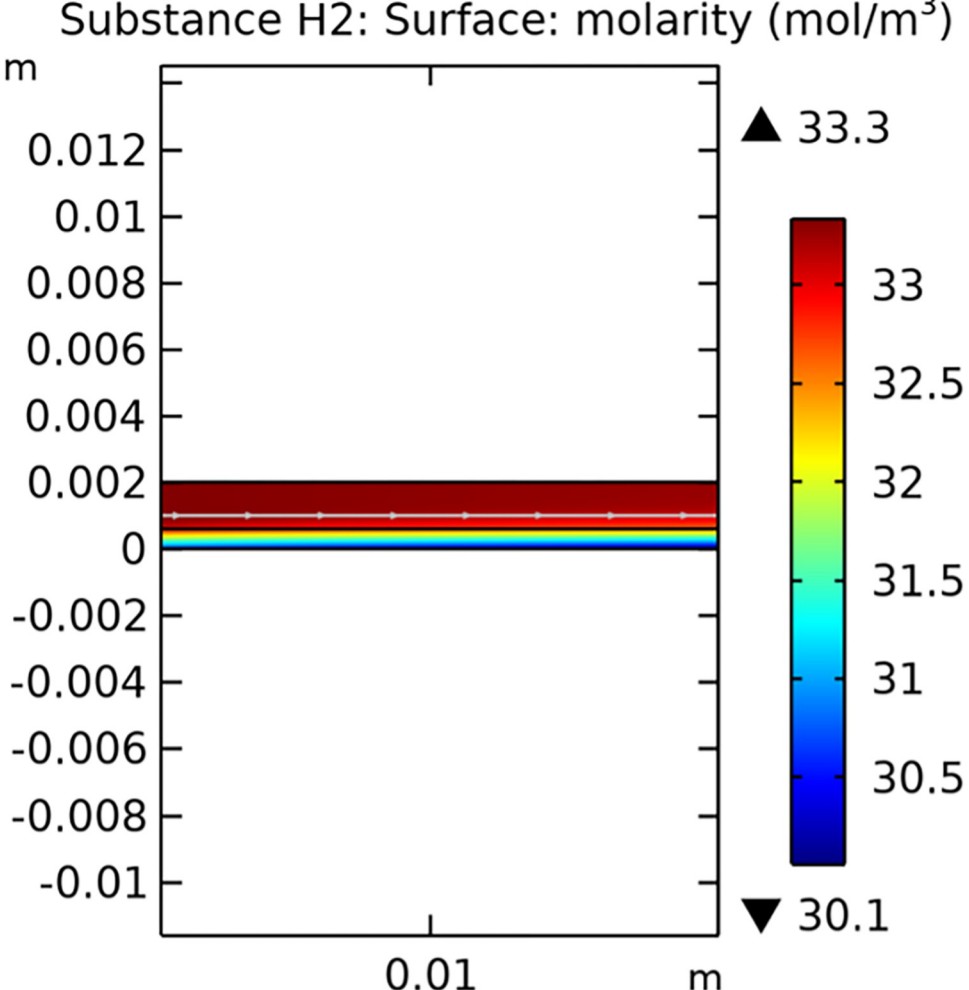

**Fig 13. Distribution of molar concentrations of hydrogen in conventional SOFC interconnector model.**

distribution plots of the conventional SOFC interconnector and the novel SOFC interconnector, it is evident that the novel interconnector causes significant disturbance in hydrogen molar fraction within the flow channel. This results in a notably higher maximum hydrogen molar fraction of 0.77 compared to the conventional SOFC interconnector. These observations underscore the novel interconnector's effectiveness in modulating hydrogen molar fraction, enhancing hydrogen transport, and promoting electrochemical reactions within the flow field channel, all of which contribute to improved SOFC performance.

Analyzing the hydrogen molar fraction distribution at an intermediate cross-section, we observe differences between the flow field and the porous anode. The porous anode shows a lower hydrogen molar fraction due to slower gas flow and diffusion rates, leading to non-uniform distribution. Conversely, the flow field channel exhibits higher hydrogen molar fraction, facilitated by faster hydrogen transport and mixing. Comparing the hydrogen molar fraction distributions at the outlet cross-section between the novel SOFC interconnector and the conventional SOFC interconnector, we observe significant differences. The novel SOFC interconnector presents a much larger gradient in hydrogen molar fraction change, which indicates a remarkable improvement in hydrogen flow rate. Moreover, the hydrogen molar fraction

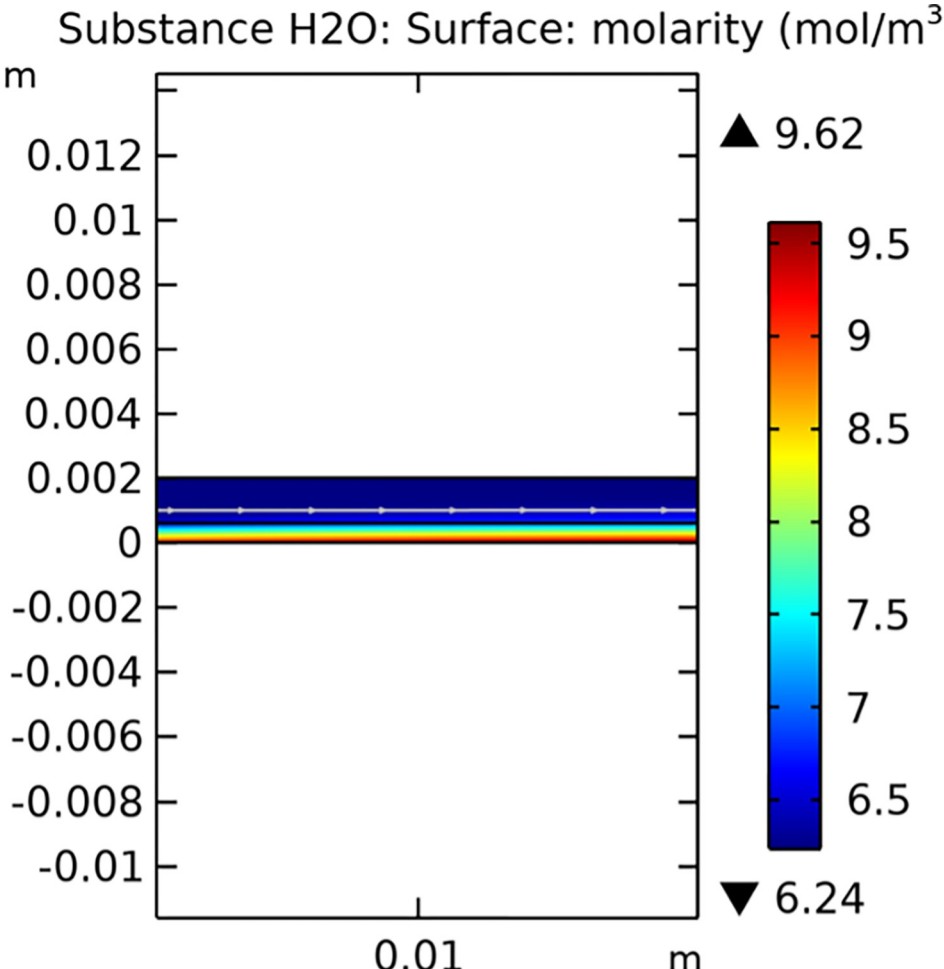

**Fig 14. Distribution of molar concentrations of water in conventional SOFC interconnector model.**

distribution in the novel SOFC interconnector at the outlet cross-section demonstrates more uniform flow and better flow rate consistency. These conclusions emphasize the advantages of the novel interconnector in hydrogen flow rate control, which achieves higher and more uniform flow rates through optimized design. This optimization is crucial for fuel cell performance, promoting efficient reactions and increased energy output.

Compared to the novel SOFC interconnector, conventional SOFC interconnector reveal distinct hydrogen distribution characteristics at the inlet. Conventional SOFC interconnector has a more uniform hydrogen concentration profile at the inlet, with a peak mole fraction. This uniformity is due to design and construction features, enabling efficient hydrogen distribution and reduced inhomogeneity. Additionally, the hydrogen molar fraction in the conventional SOFC interconnector remains stable at 0.8, which ensures consistent high hydrogen supply and improved performance. The hydrogen distribution in the novel interconnector SOFC shows a distinctive jump discontinuity at the porous anode-fluid channel interface due to ribs. This leads to a higher average hydrogen concentration in the novel SOFC interconnector compared to conventional ones, potentially improving performance. Further optimization of ribs can enhance hydrogen distribution uniformity and overall SOFC efficiency.

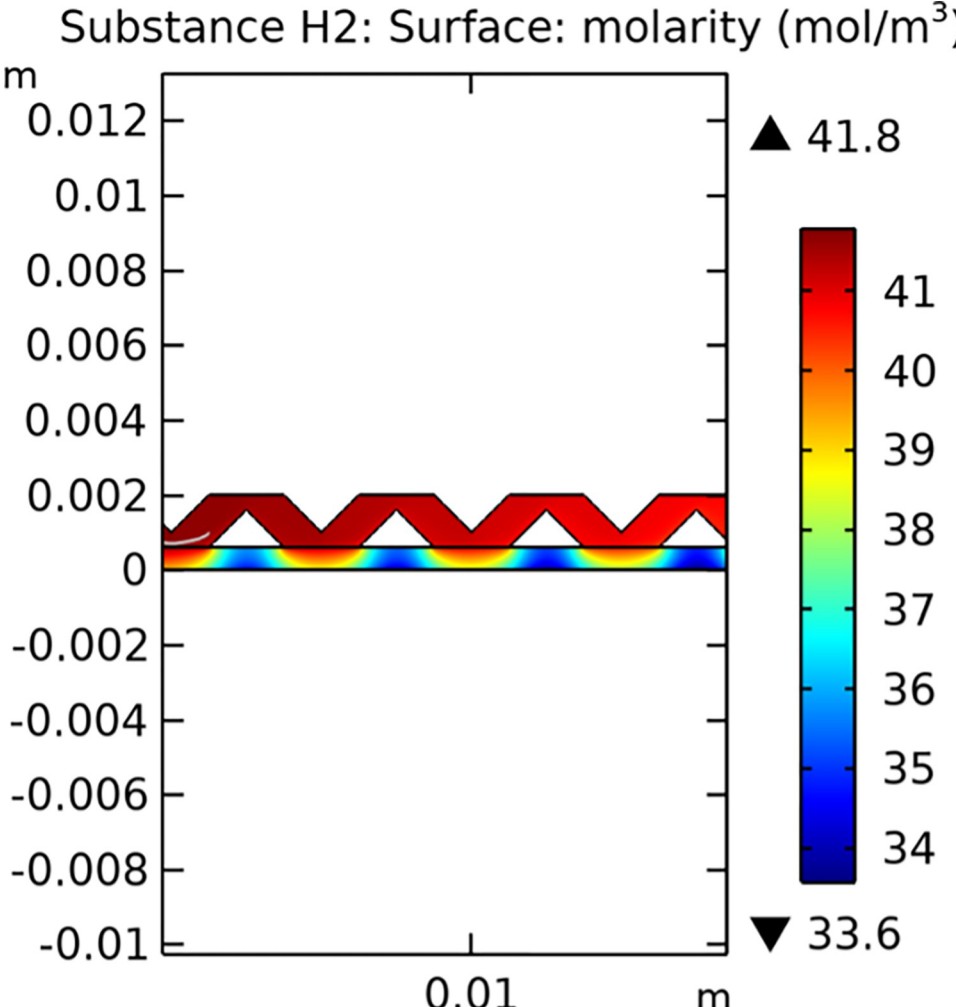

**Fig 15. Distribution of molar concentrations of hydrogen in novel SOFC interconnector model.**

In comparing the Figs 29–31, smaller rib heights had minimal impact on hydrogen flow, similar to the 2D model with a conventional interconnector. However, as rib height increased beyond a certain point, they began to positively influence the flow field, significantly enhancing hydrogen flow rates. This enhancement is crucial for cell performance. Therefore, thoughtful rib design can improve hydrogen distribution and efficiency, but it is essential to carefully consider and balance rib height selection to avoid hindering hydrogen flow. Based on Figs 32–34, the rib arrangement significantly affects hydrogen pressure. Ribs raise localized pressure in the fluid channel, increasing hydrogen flow speed. With smaller rib heights, pressure appears dark blue in the cloud diagram, gradually transitioning to light blue and even yellow at the inlet as rib height increases. This indicates that rib arrangements enhance the flow field and average pressure within the porous anode, benefiting the cell. Within structural strength limits, this pressure increase offers several advantages. It promotes efficient hydrogen transfer by facilitating faster flow through channels and the porous anode. Additionally, it improves hydrogen distribution within the cell, ensuring even access to the reaction area, enhancing cell uniformity, and performance, which is depicted in Figs 44–46.

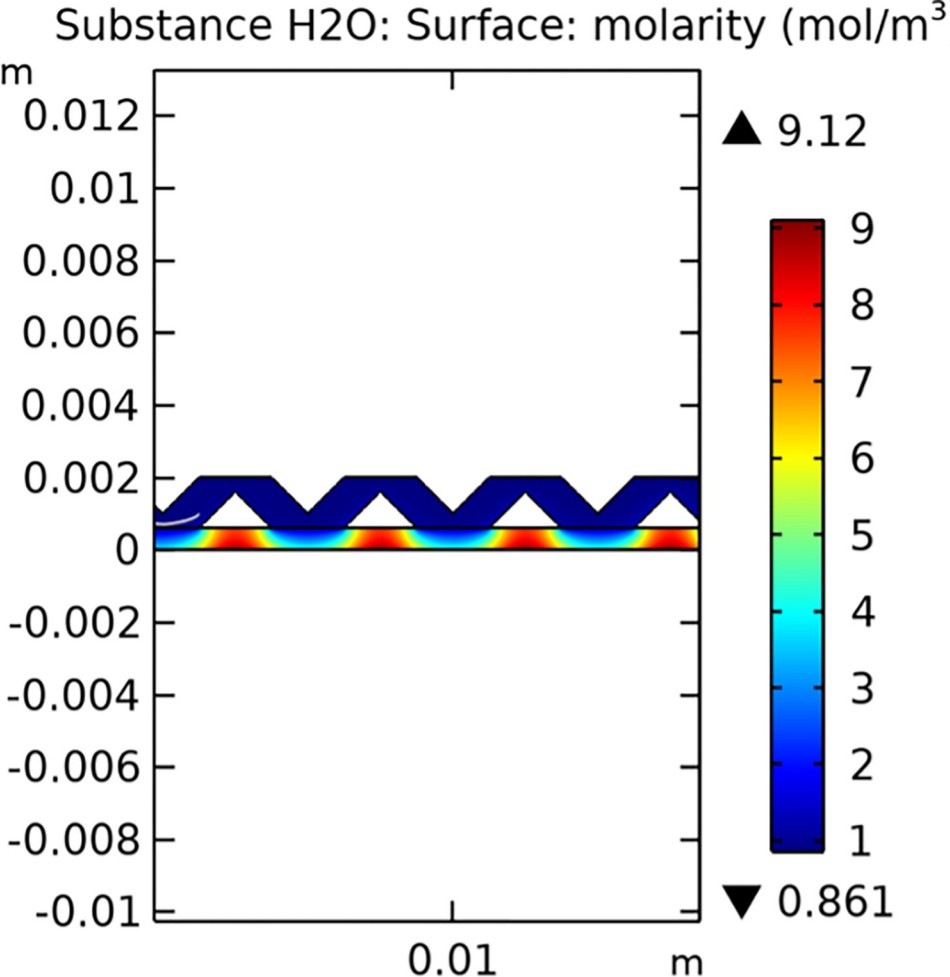

**Fig 16. Distribution of molar concentrations of water in novel SOFC interconnector model.**

The effectiveness of rib arrangements depends on their dimensional structure. When the sum of the heights of the two rib layers is less than the width of the fluid channel, the impact on hydrogen concentration is minimal through comparing Figs 35–37. However, when this sum exceeds the channel width, rib height directly influences hydrogen concentration. Increasing rib height perturbs hydrogen concentration, benefiting cell performance, but excessive height obstructs fluid flow, potentially harming the model. According to Figs 41–43, we also make this conclusion. Thus, rib height selection plays a crucial role in hydrogen concentration distribution and cell performance. A balanced rib height is essential to optimize both factors in interconnector design.

In Figs 38–40, changes in interconnector structure have limited impact on the porous anode's interior. This is because the porous anode's dense structure minimizes direct rib arrangement effects. Moreover, the slow hydrogen flow rate within the porous anode is less influenced by interconnector structural changes. The high porosity and surface area of the porous anode create a complex hydrogen diffusion path, primarily determined by the anode's structure and pore distribution. While interconnector structure alterations may slightly affect hydrogen diffusion paths within the anode, the porous anode's inherent high porosity and surface area minimize this impact. Furthermore, the hydrogen flow rate within the porous anode

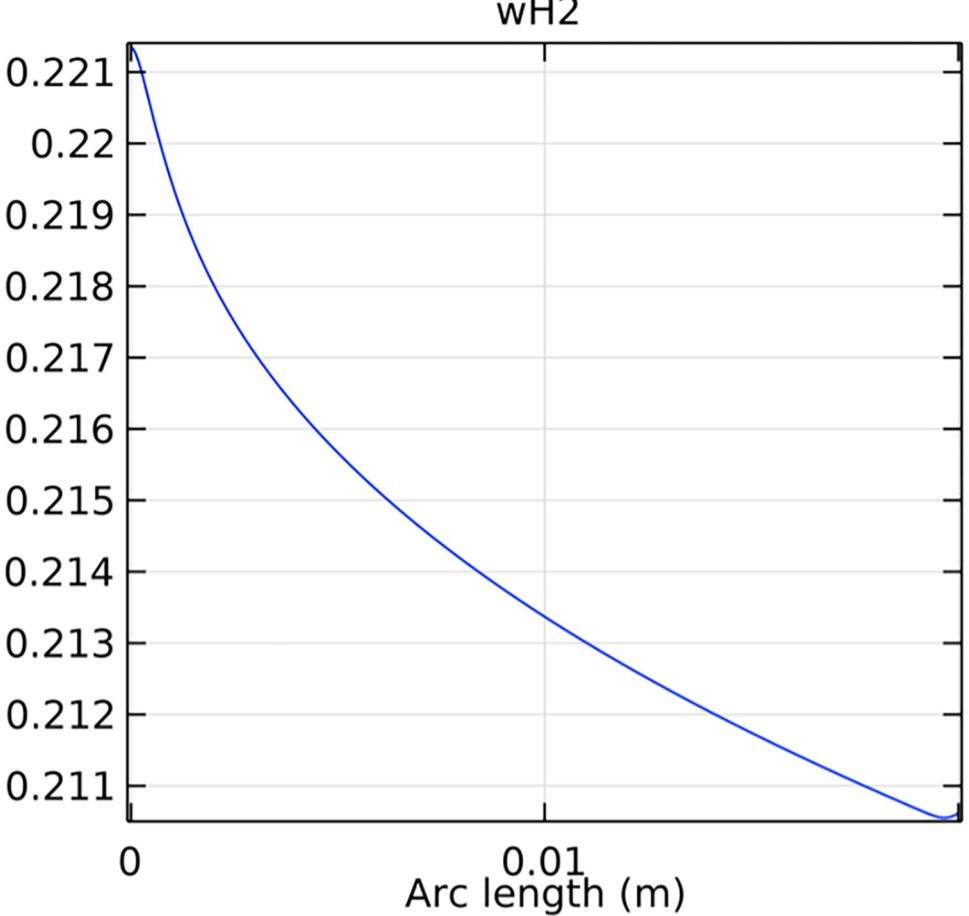

**Fig 17. Hydrogen molar fraction distribution within the conventional SOFC porous anode.**

is predominantly governed by hydrogen diffusion in the pores, making interconnector structural changes generally have minimal effects.

## 5. Conclusions

This study investigated the influence of interconnector structures on the flow and mass transfer characteristics of SOFCs through comparative and optimization analyses. The novel interconnector structure demonstrated advantages in terms of flow characteristics, enhancing gas transport efficiency and uniformity. However, it also showed localized molar fraction changes in the porous anode. Additionally, different ribs had varying impacts on flow and hydrogen transport, affecting key parameters like hydrogen mole fraction, flow rate, and pressure. Optimizing interconnector structural parameters holds significant potential for improving SOFC performance and efficiency.

While this research provides valuable insights into SOFC design optimization, there are certain limitations to address in future work:

1. Three-dimensional simulations should be employed when necessary to model complex interconnector structures, introducing additional physical fields to enhance design optimization.

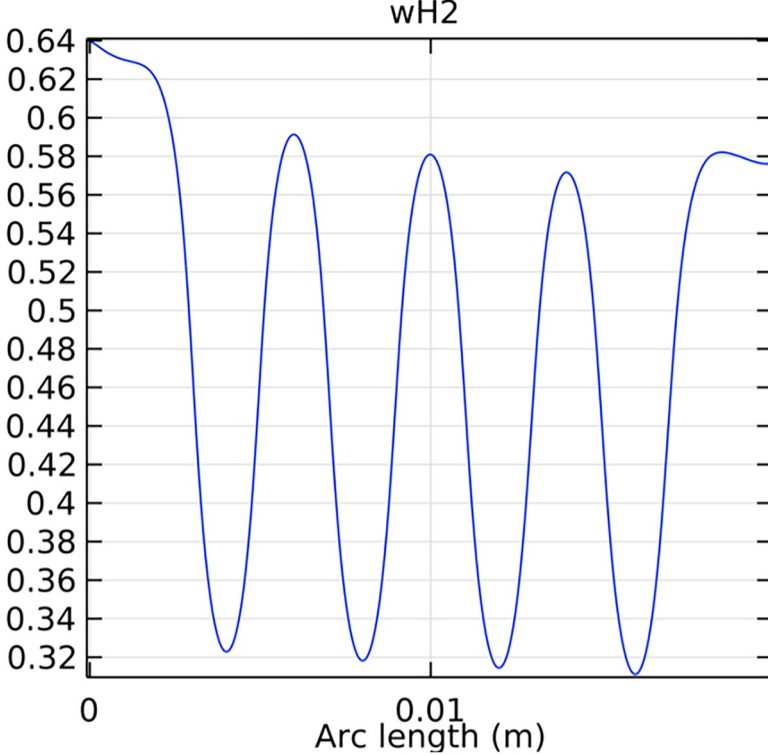

**Fig 18. Hydrogen molar fraction distribution within the novel SOFC porous anode.**

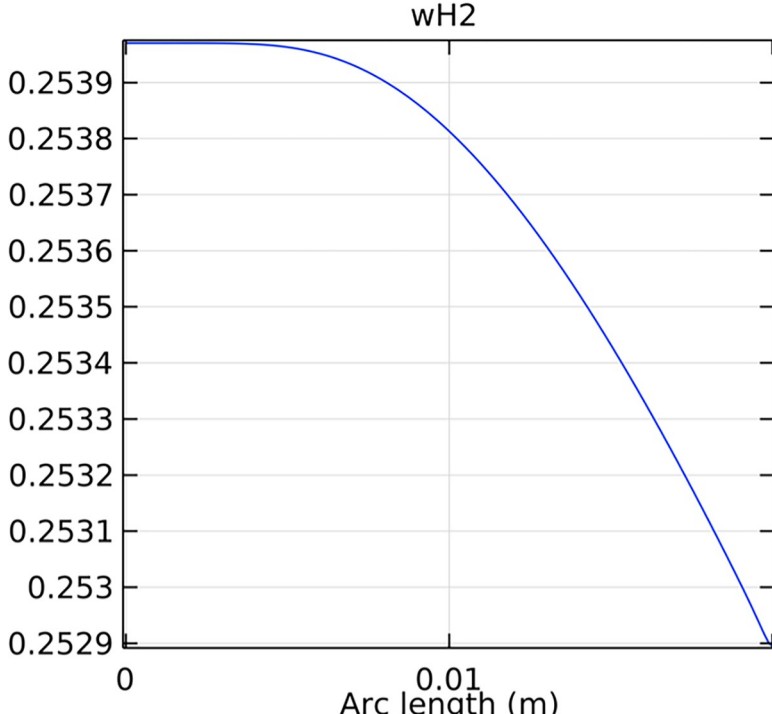

**Fig 19. Hydrogen molar fraction distribution within the flow field channel of the conventional interconnector SOFC.**

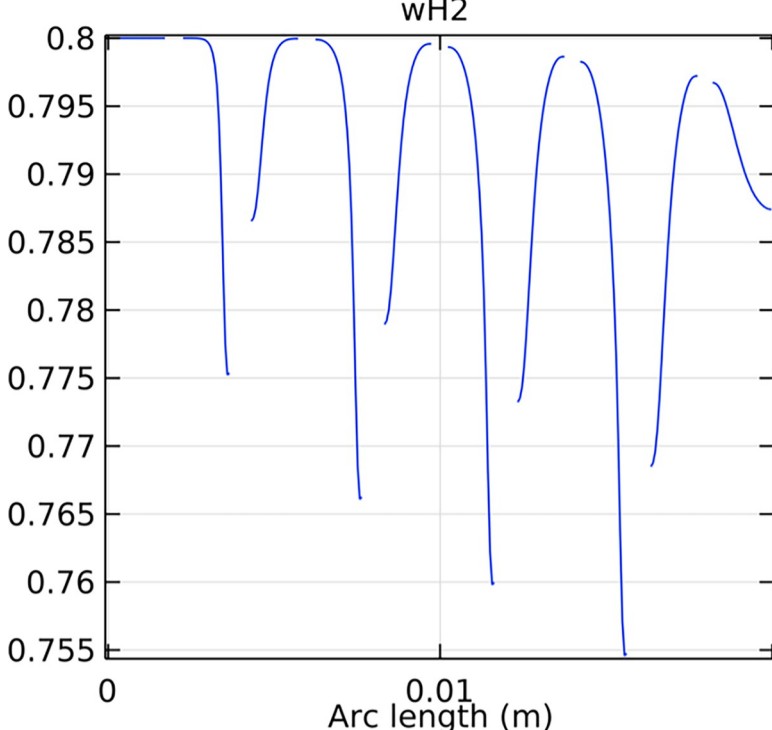

**Fig 20. Hydrogen molar fraction distribution within the flow field channel of the novel interconnector SOFC.**

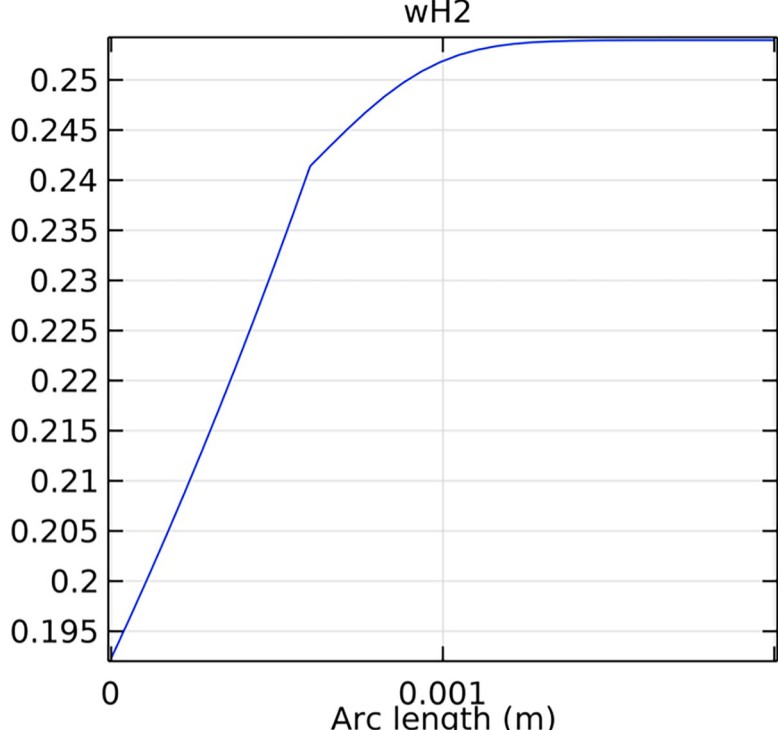

**Fig 21. Hydrogen molar fraction distribution of the conventional interconnector SOFC at intermediate cross section.**

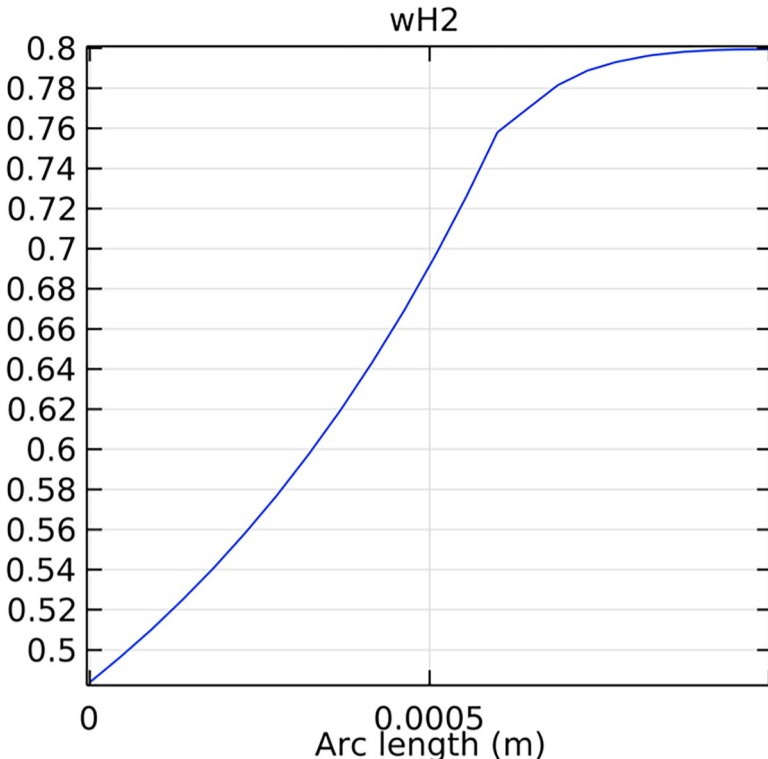

**Fig 22. Hydrogen molar fraction distribution of the novel interconnector SOFC at intermediate cross section.**

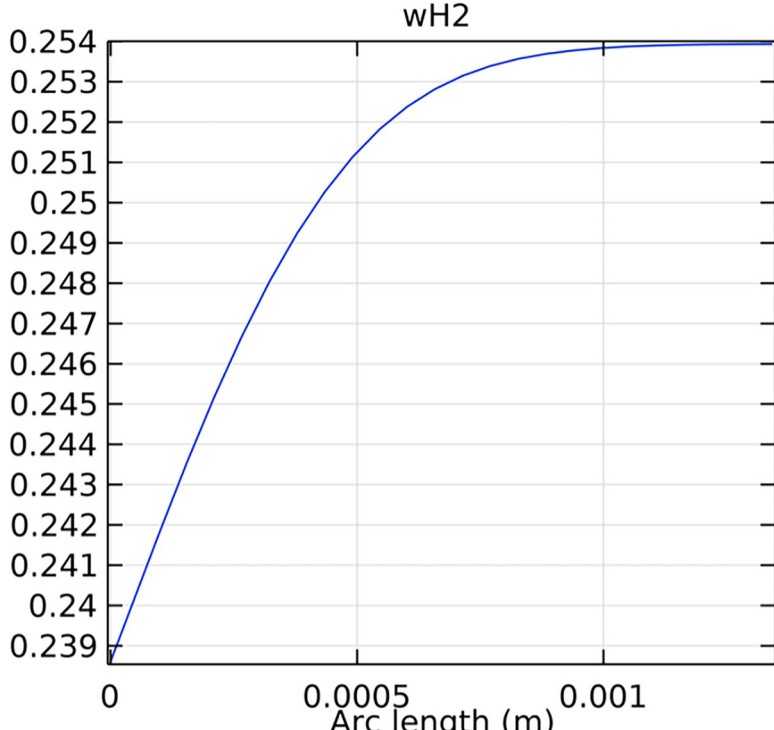

**Fig 23. Hydrogen molar fraction distribution at the outlet cross-section of the conventional SOFC interconnector.**

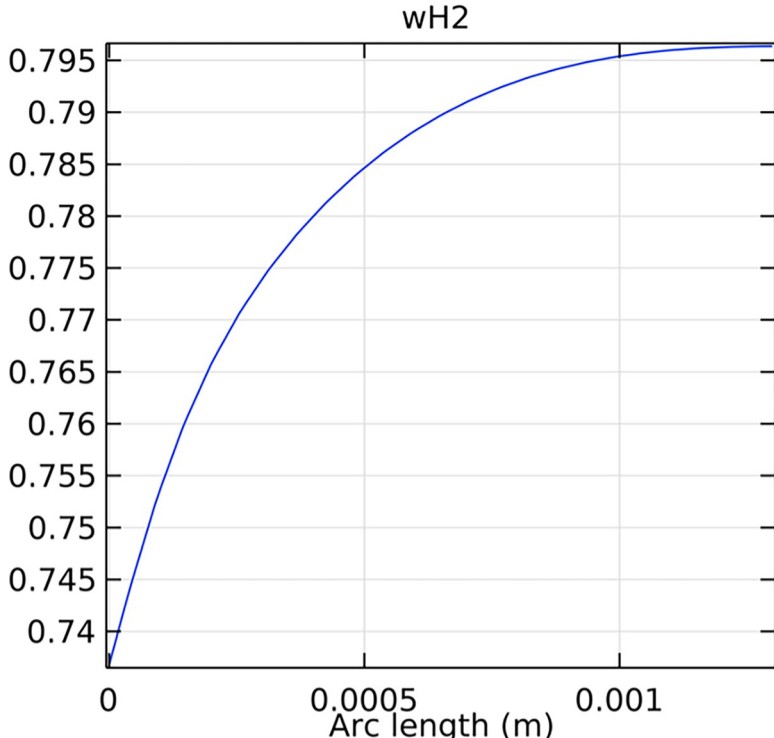

**Fig 24. Hydrogen molar fraction distribution of the outlet cross-section of the novel SOFC interconnector.**

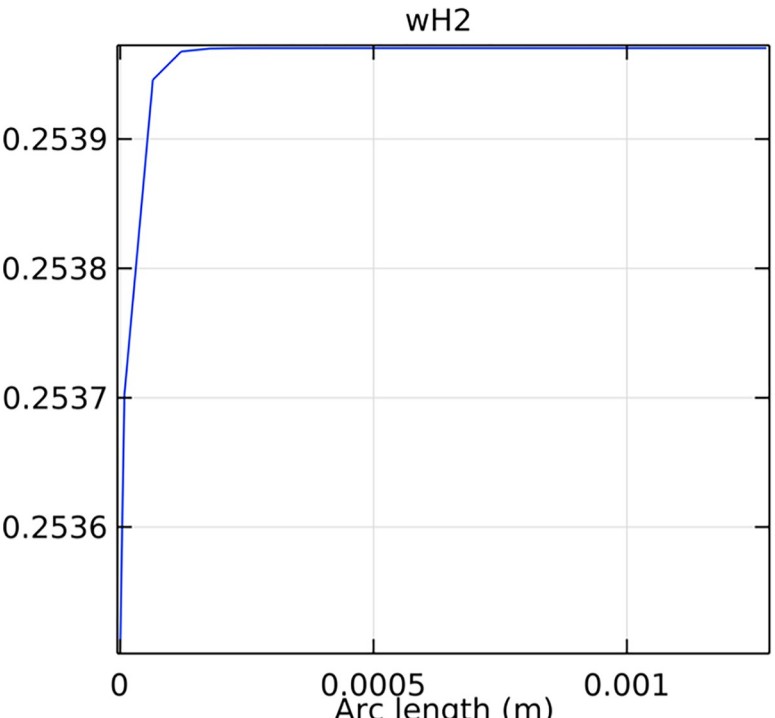

**Fig 25. Hydrogen molar fraction distribution in the inlet cross-section of the conventional SOFC interconnector.**

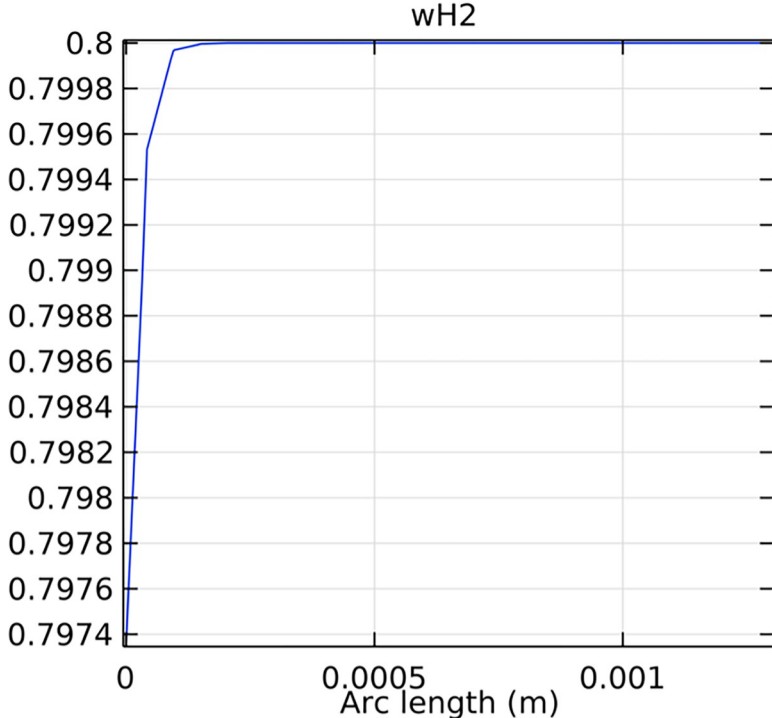

**Fig 26. Hydrogen molar fraction distribution of the inlet cross-section of the novel SOFC interconnector.**

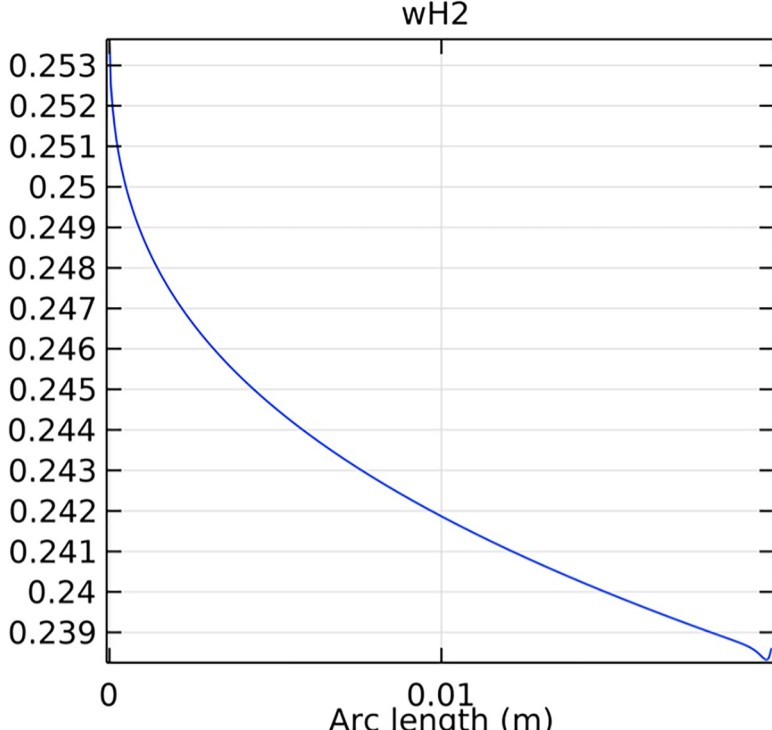

**Fig 27. Hydrogen molar fraction distribution at the interface of the conventional SOFC interconnector porous anode.**

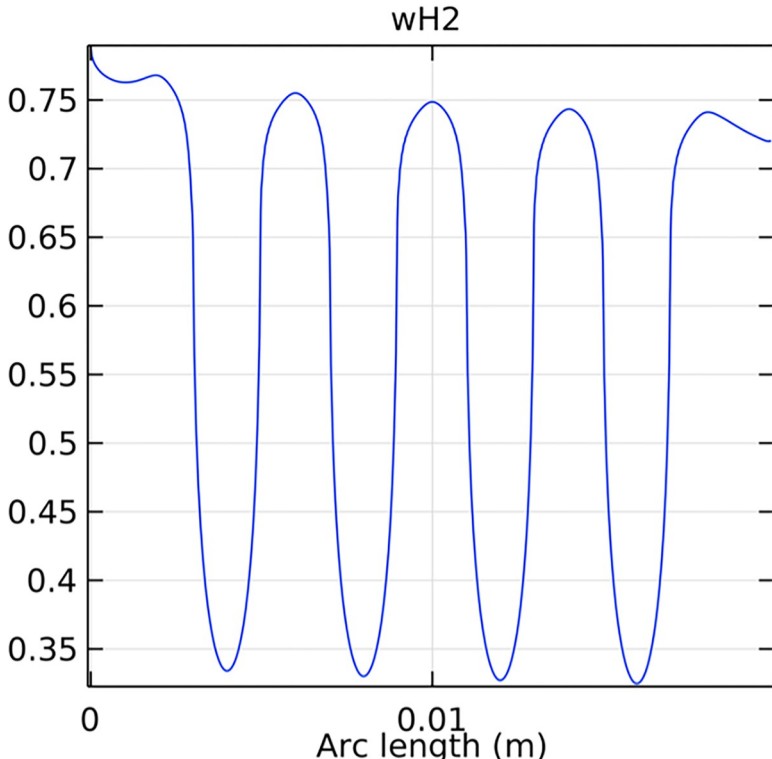

**Fig 28. Hydrogen molar fraction distribution at the interface of the novel SOFC interconnector porous anode.**

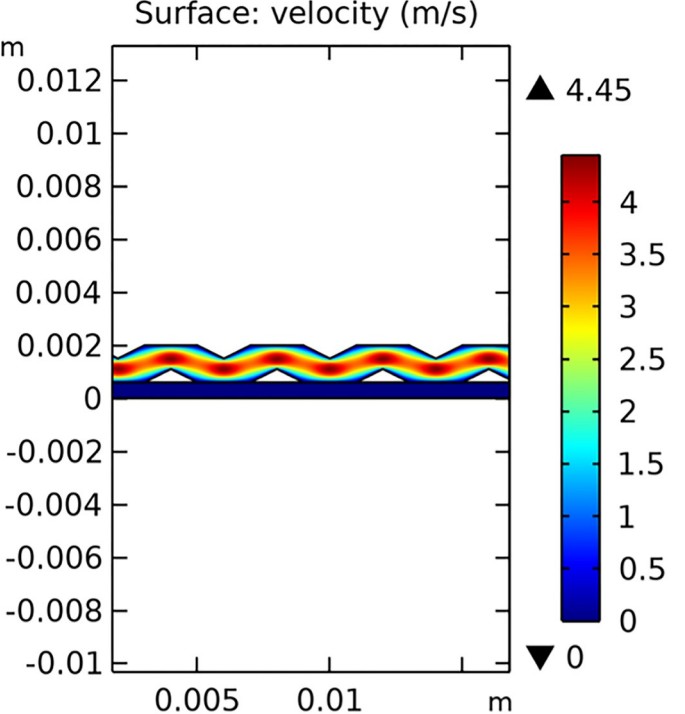

**Fig 29. Hydrogen flow rate distribution of the novel SOFC interconnector type I.**

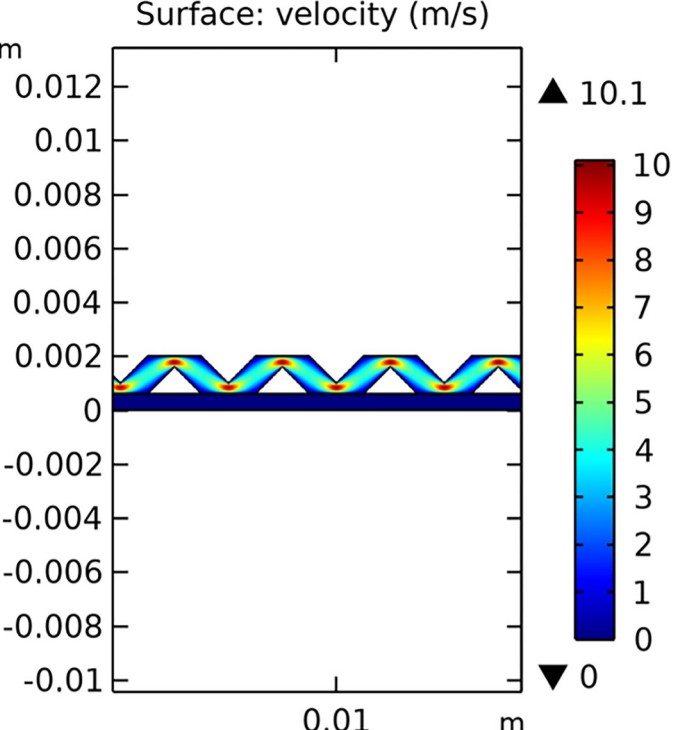

**Fig 30. Hydrogen flow rate distribution of the novel SOFC interconnector type II.**

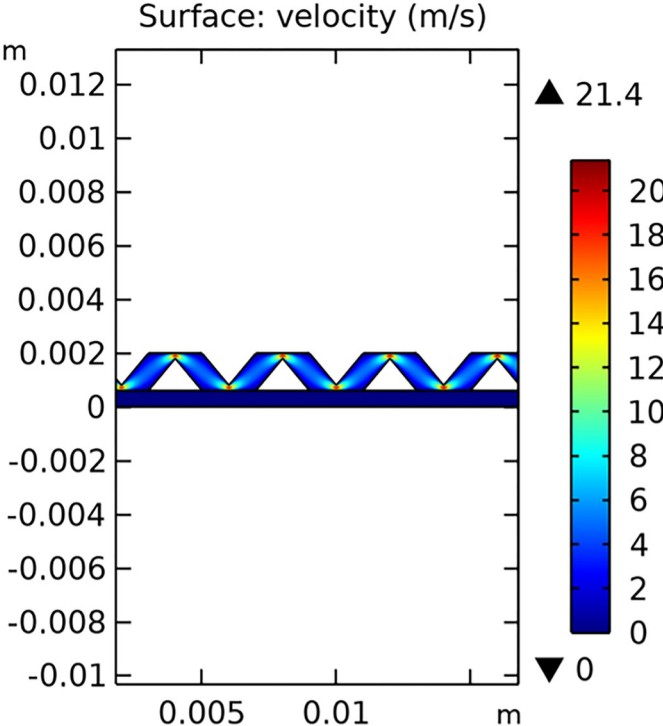

**Fig 31. Hydrogen flow rate distribution of the novel SOFC interconnector type III.**

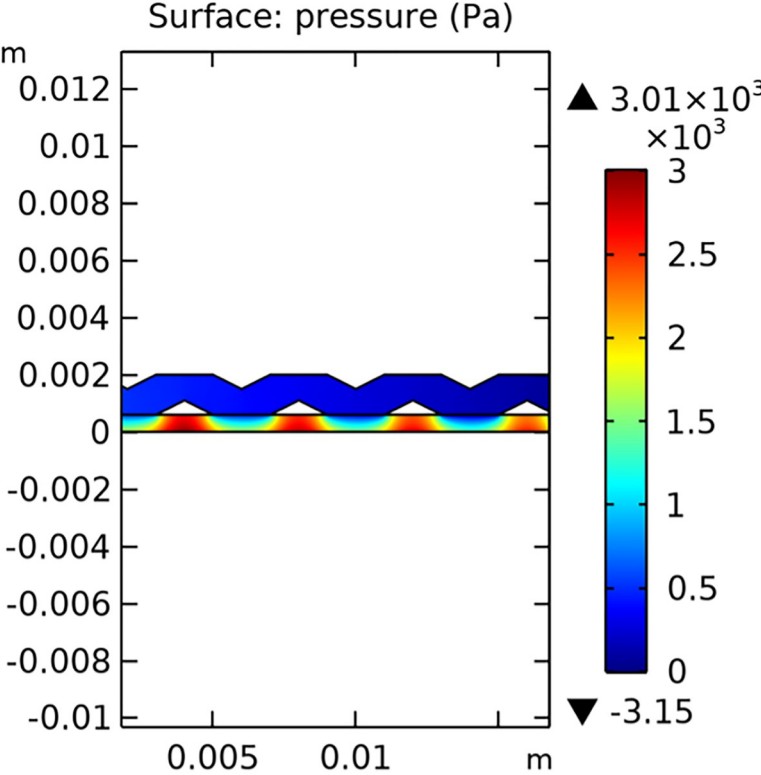

**Fig 32. Pressure distribution of the novel SOFC interconnector type I.**

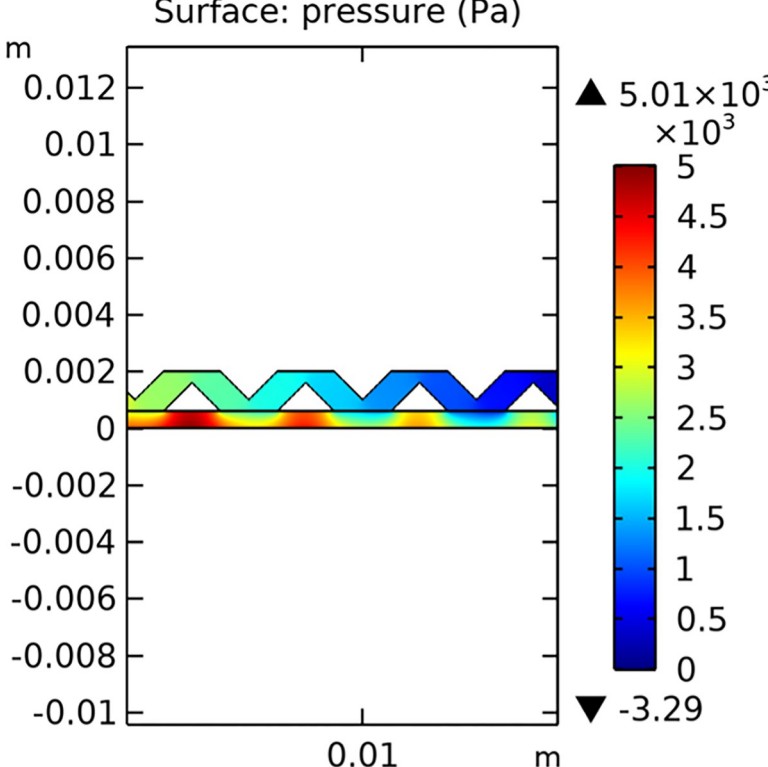

**Fig 33. Pressure distribution of the novel SOFC interconnector type II.**

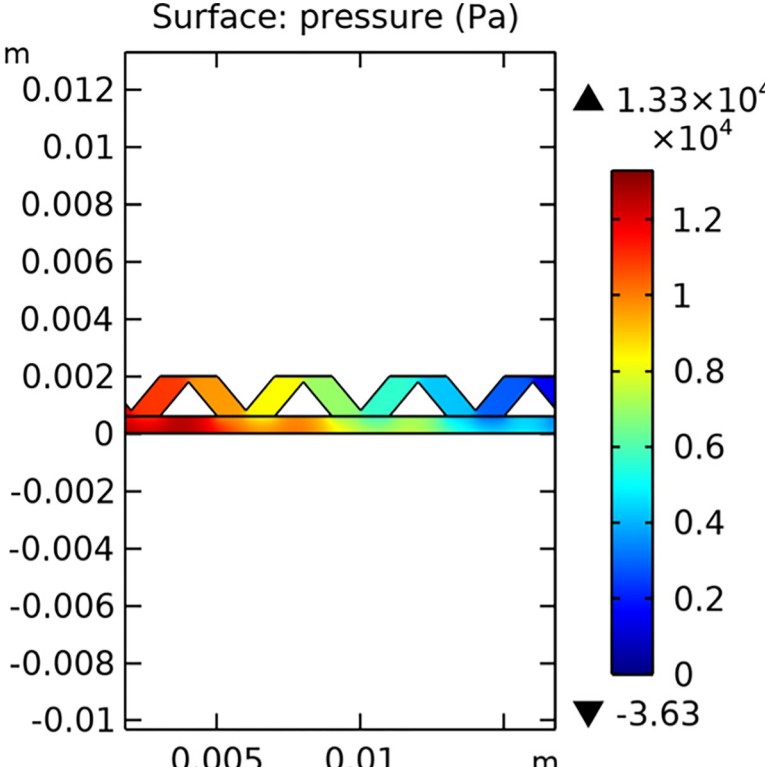

**Fig 34. Pressure distribution of the novel SOFC interconnector type III.**

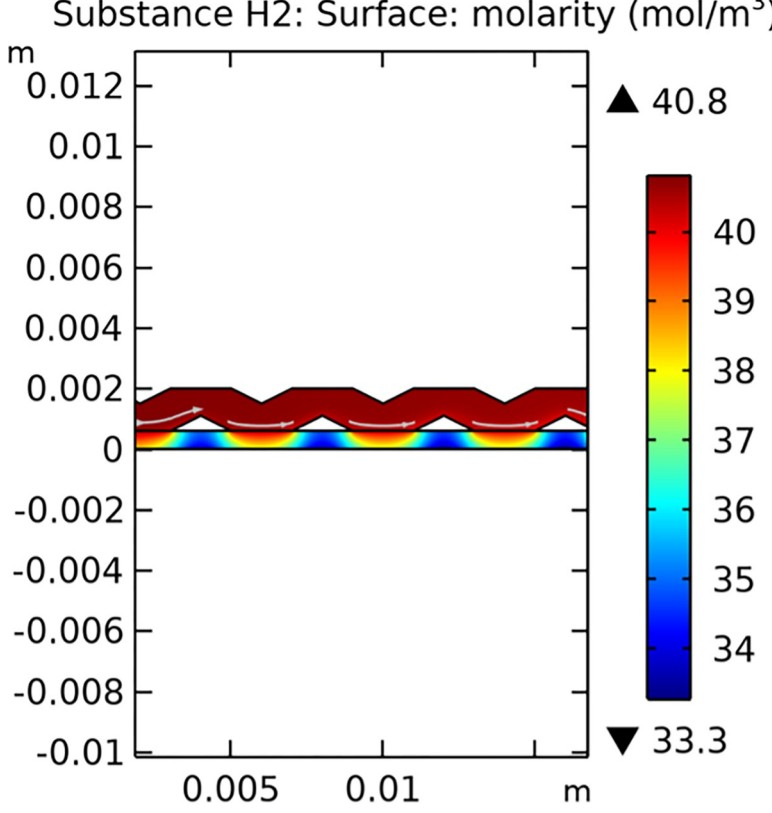

**Fig 35. Hydrogen concentration distribution of the novel SOFC interconnector type I.**

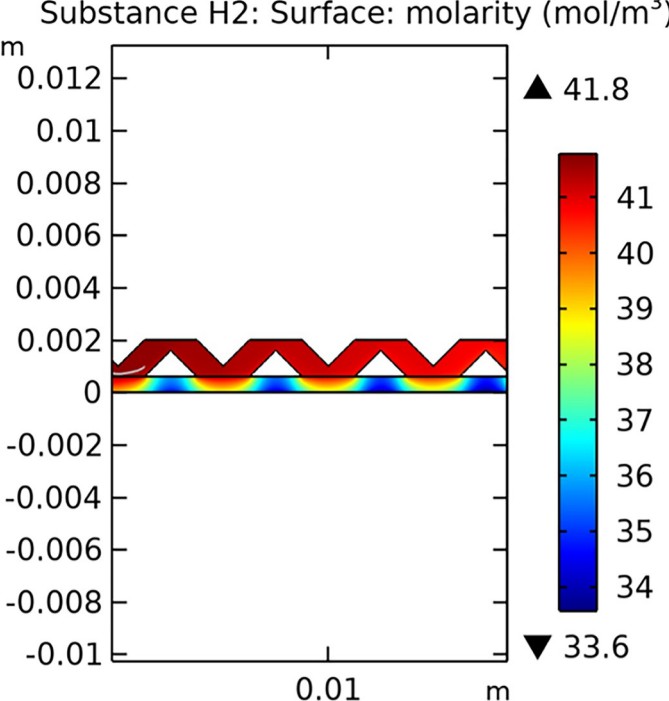

**Fig 36. Hydrogen concentration distribution of the novel SOFC interconnector type II.**

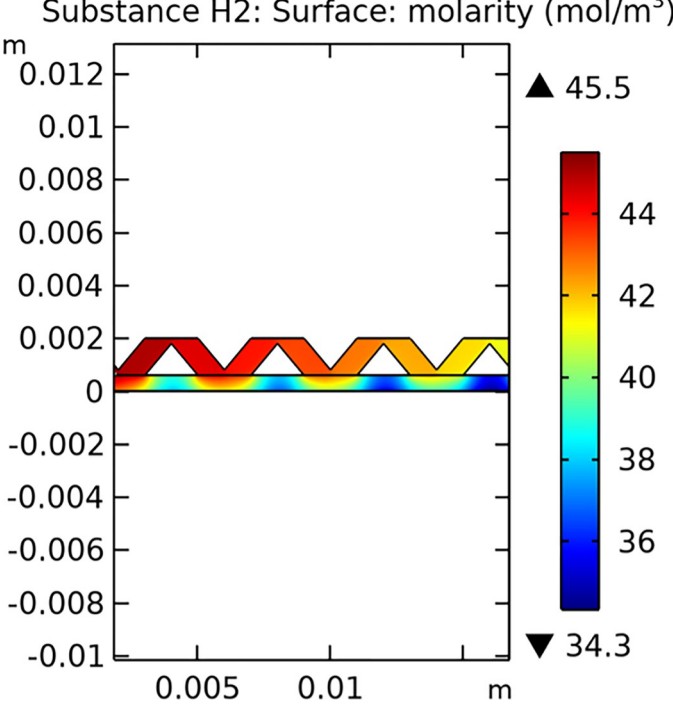

**Fig 37. Hydrogen concentration distribution of the novel SOFC interconnector type III.**

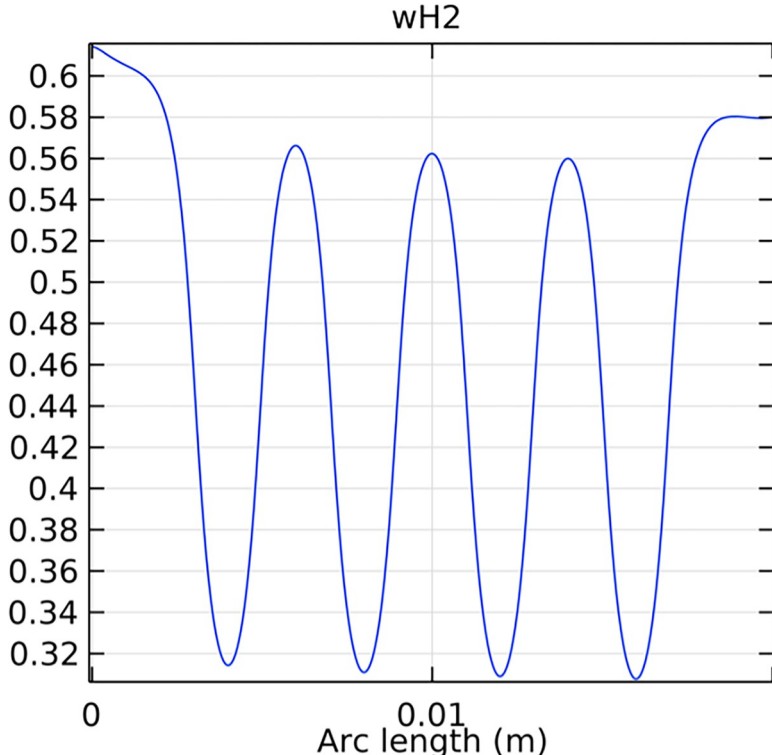

**Fig 38. Hydrogen molar fraction distribution of the novel SOFC interconnector porous anode type I.**

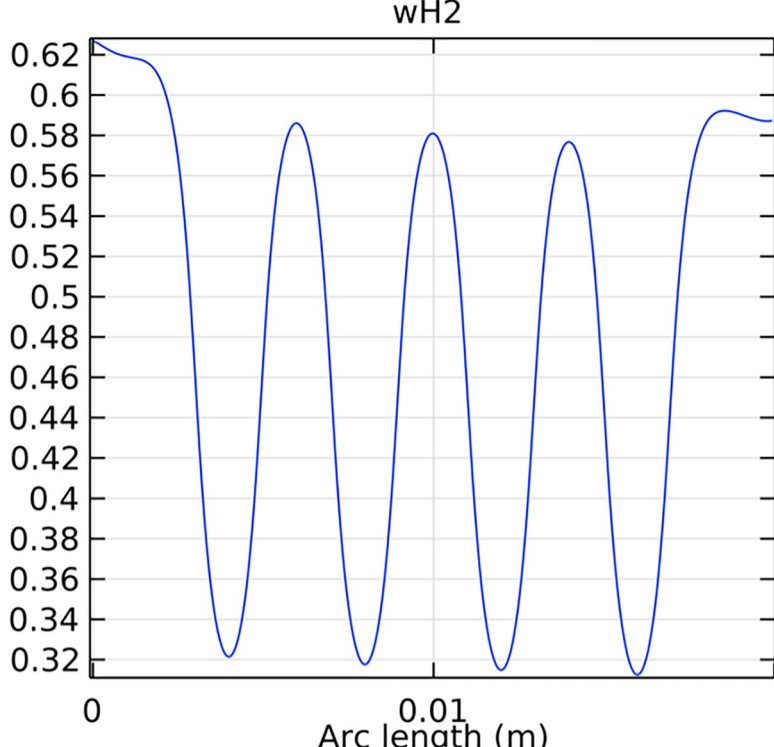

**Fig 39. Hydrogen molar fraction distribution of the novel SOFC interconnector porous anode type II.**

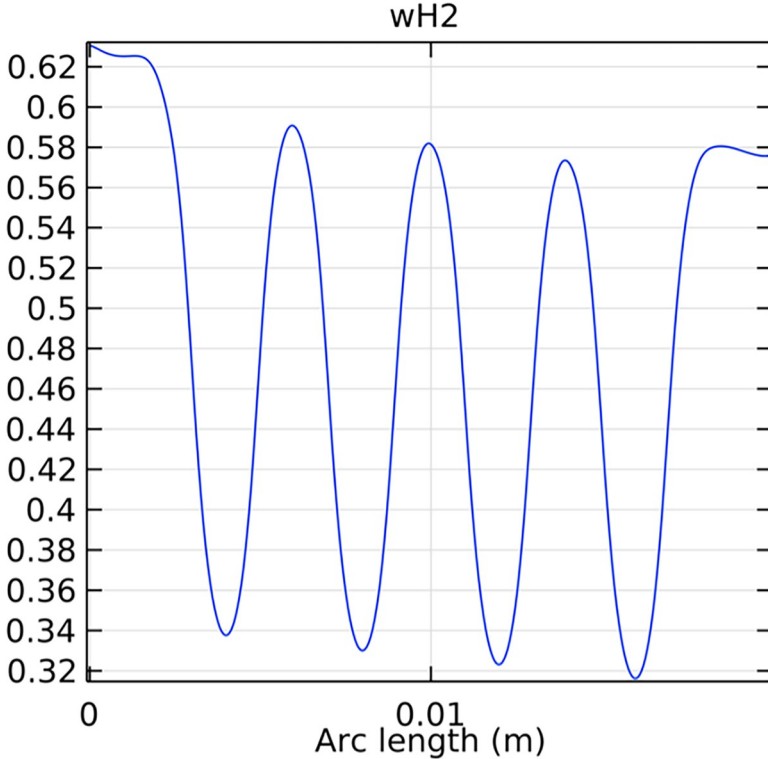

**Fig 40. Hydrogen molar fraction distribution of the novel SOFC interconnector porous anode type III.**

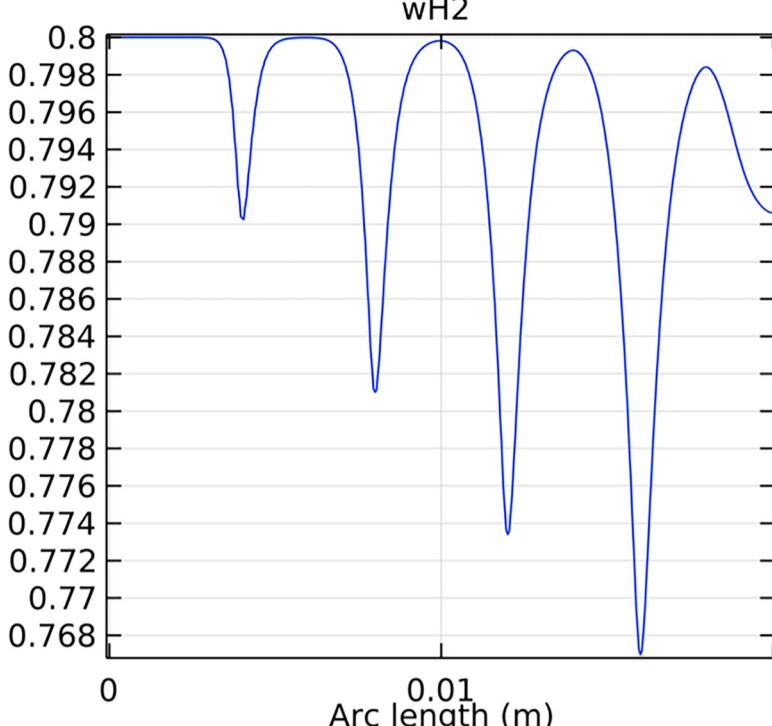

**Fig 41. Hydrogen concentration distribution in the fluid channel of the novel SOFC interconnector type I.**

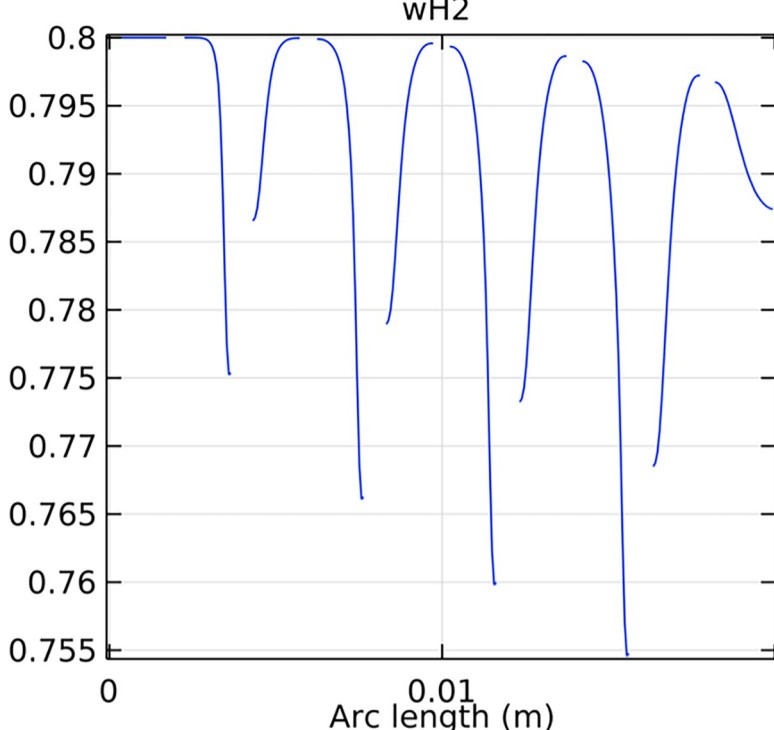

**Fig 42. Hydrogen concentration distribution in the fluid channel of the novel SOFC interconnector type II.**

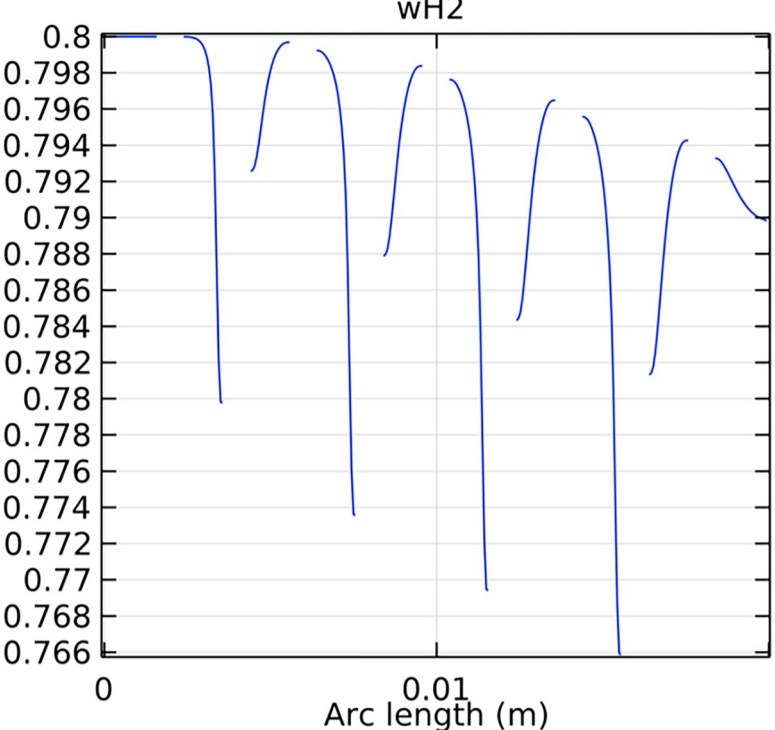

**Fig 43. Hydrogen concentration distribution in the fluid channel of the novel SOFC interconnector type III.**

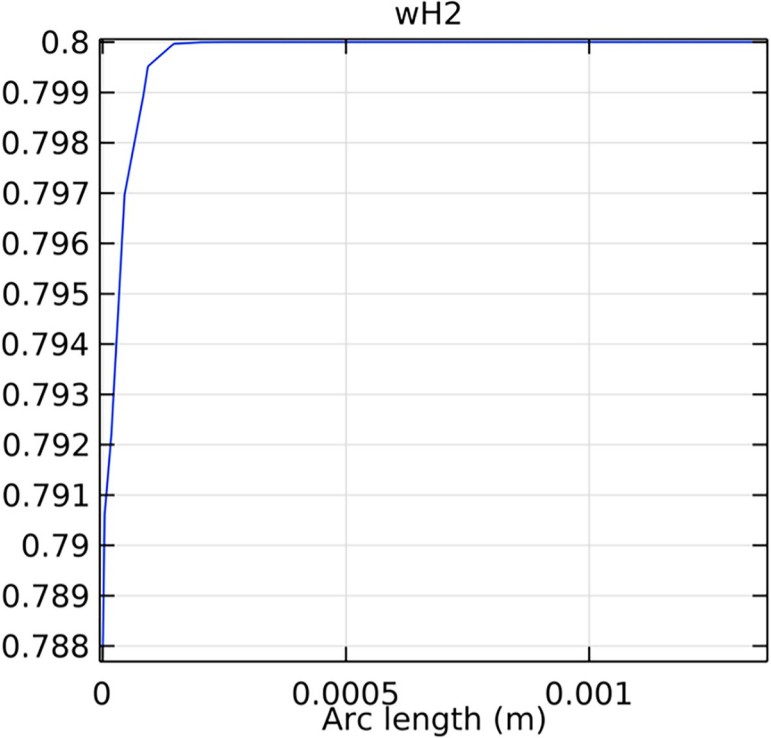

**Fig 44. Distribution of hydrogen concentration in the inlet cross-section of the novel SOFC interconnector type I.**

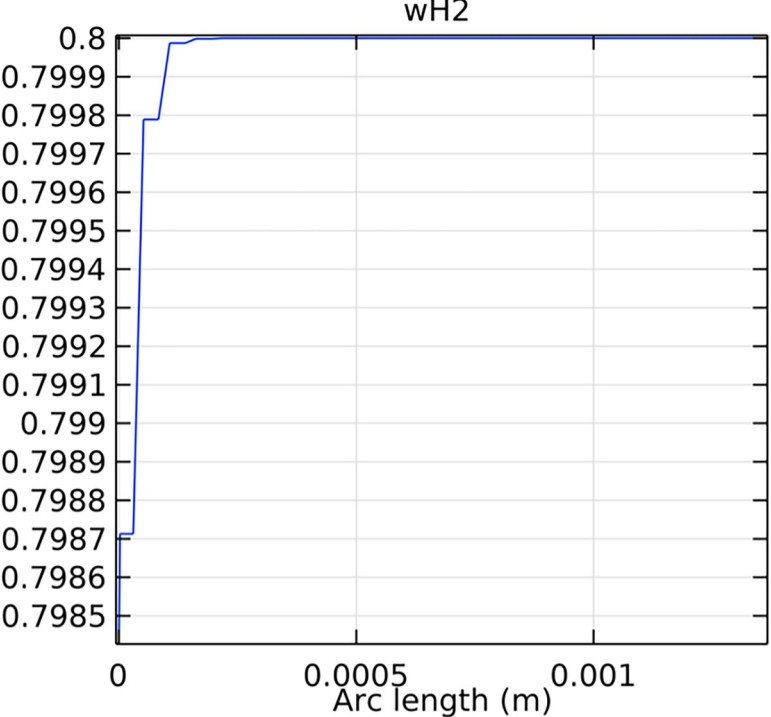

**Fig 45. Distribution of hydrogen concentration in the inlet cross-section of the novel SOFC interconnector type II.**

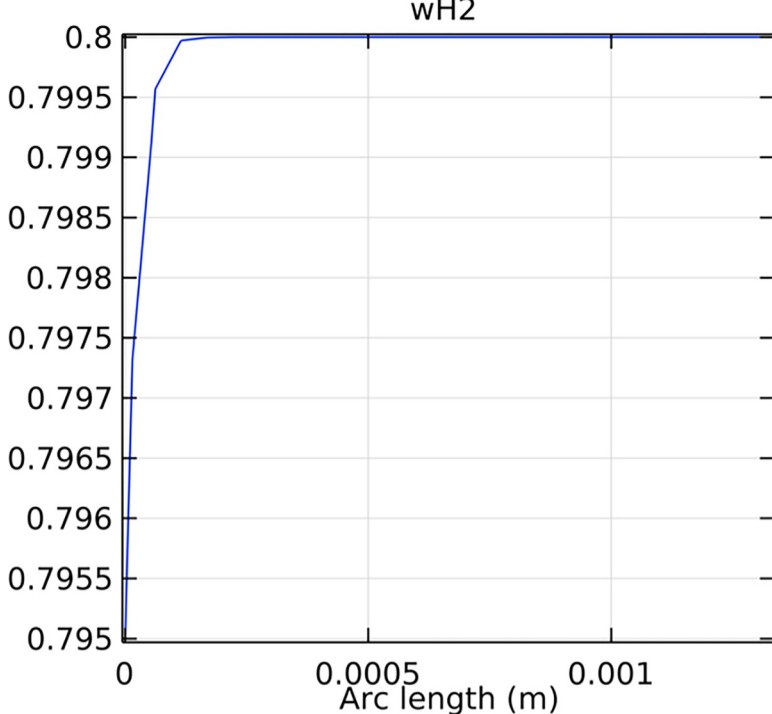

**Fig 46. Distribution of hydrogen concentration in the inlet cross-section of the novel SOFC interconnector type III.**

2. Future studies should include an investigation of electrical properties and consider inter-layer and intralayer gas flow for a more comprehensive analysis.

3. The study of high-temperature oxidation and its impact on interconnector performance over time is an important dynamic process that should be considered in future research.

The field of fuel cell technology offers substantial room for performance improvement, and this work contributes to our understanding of interconnector structure effects. Further research can explore different interconnector structure designs and experimentally validate numerical simulation results.

## Acknowledgments

We are very grateful to the following people who contributed to the field investigations and laboratory work: Pei Fu.

## Author Contributions

**Conceptualization:** Boxiang Sun, Xiang Shao.

**Data curation:** Boxiang Sun, Xiang Shao.

**Formal analysis:** Boxiang Sun.

**Funding acquisition:** Xiang Shao.

**Investigation:** Xiang Shao.

**Methodology:** Huiyu Wang, Songyan Zou.

**Project administration:** Xiang Shao.

**Resources:** Boxiang Sun.

**Software:** Boxiang Sun.

**Writing – original draft:** Boxiang Sun.

**Writing – review & editing:** Xiang Shao.

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
