## [Decision Letter · Decision Letter 0]

27 Nov 2023

PONE-D-23-31430Optimized design of planar solid oxide fuel cell interconnectorsPLOS ONE

Dear Dr. Shao,

Thank you for submitting your manuscript to PLOS ONE. After careful consideration, we feel that it has merit but does not fully meet PLOS ONE’s publication criteria as it currently stands. Therefore, we invite you to submit a revised version of the manuscript that addresses the points raised during the review process.

**ACADEMIC EDITOR: **The reviewers recommend reconsideration the manuscript with revision and modification. I invite the authors to resubmit the manuscript after addressing the comments raised by the reviewers.

We look forward to receiving your revised manuscript.

Kind regards,

Dhanamjayulu C, Ph.D & Post.Doc

Academic Editor

PLOS ONE

Journal Requirements:

"The study was financially supported by to Postgraduate Research & Practice Innovation Program of Jiangsu Province (Grant No. SJCX22_1067)."

Additional Editor Comments:

The reviewers recommend reconsideration the manuscript with revision and modification. I invite the authors to resubmit the manuscript after addressing the comments raised by the reviewers.

Reviewers' comments:

Reviewer's Responses to Questions

**Comments to the Author**

1. Is the manuscript technically sound, and do the data support the conclusions?

Reviewer #1: Partly

Reviewer #2: Yes

2. Has the statistical analysis been performed appropriately and rigorously? 

Reviewer #1: Yes

Reviewer #2: N/A

3. Have the authors made all data underlying the findings in their manuscript fully available?

Reviewer #1: Yes

Reviewer #2: Yes

4. Is the manuscript presented in an intelligible fashion and written in standard English?

Reviewer #1: Yes

Reviewer #2: No

5. Review Comments to the Author

Reviewer #1: The authors present a paper on the design of interconnects for planar Solid Oxide Fuel Cells geometry. Such topic is very important for the SOFC technology since the Interconnect is a critical component that has to ensure optimal gas distribution and electrical connection in order to prevent additional losses in efficiency of the system.

This work is providing 2D simulation of several parameters as the gas velocity, pressure, and hydrogen molar fraction distribution.

The setup of the model is sound, however the manuscript lacks of several details as listed below:

1) Authors compare the calculations performed on a “conventional” interconnect and a “novel” interconnect design, however the “conventional” design is just a simple geometry that is actually not in use for real systems. The “novel” design is something closer to the state-of-art interconnect design being characterized by the presence of channels with the function of distributing gases and contacts point aiming at collecting electrical currents. Even neglecting the feasibility of the proposed “novel” design, characterized by sharp edges and abrupt variations in geometry, which is not compatible with common manufacturing methods, I would expect a study on the electrical contribution too. The effect of geometry on the current collection should not be neglected. In fact, authors state this in the Section 1.3 but such aspect is not provided.

2) Figure 3.1 is the same as Figure 2.3, I’m not sure is necessary to provide the same figure two times. It would be rather appropriate, for the sake of clarity, to add text boxes into the figures indicating the position and details of the sketch (for example:, rib, channel, electrode…). Also, the flow direction should be indicated by an arrow in the sketches, or at least an indication of the inlet and outlet positions.

3) In figures 4.1, 4.2, 4.3, to highlight the differences in the designs it would be helpful to indicate the quotes that are changing. It would be also helpful to indicate where the interconnect is in contact with the electrode and with the gas flow.

4) Several graphs lack the axis labels.

Reviewer #2: The findings and results in this study are meaningful.

The article provides an idea for a novel 2D simulation model for interconnector SOFCs, aiming to enhance

their performance. The findings and results in this study are meaningful. The introduction, results and conclusions of the manuscript are clear and well-written from the reviewer's perspective.

Although the authors needs point out more the differences between this article and previous research in the introduction, also the authors should reduce the similarity rate of the paper which is 37% in the current version.

6. PLOS authors have the option to publish the peer review history of their article (what does this mean?). If published, this will include your full peer review and any attached files.

Reviewer #1: No

Reviewer #2: **Yes: **Ali Majdi

---

## [Author Response · Author response to Decision Letter 0]

13 Jan 2024

Response letter

Dear editor,

Many thanks to you and the reviewer for your comments and suggestions for our manuscript (Submission ID: PONE-D-23-31430). According to the comments and suggestions, we have made a series of revisions on the manuscript. We used the "Track Changes" function to mark up our revisions. Please see the revised manuscript. The follows are point-to-point responses to the comments and suggestions.

Thank you very much and best wishes. 

Yours sincerely, 

Boxiang Sun (the first author)

Xiang Shao (the corresponding author)

PONE-D-23-31430

Optimized design of planar solid oxide fuel cell interconnectors

PLOS ONE

Dear Dr. Shao,

Thank you for submitting your manuscript to PLOS ONE. After careful consideration, we feel that it has merit but does not fully meet PLOS ONE’s publication criteria as it currently stands. Therefore, we invite you to submit a revised version of the manuscript that addresses the points raised during the review process.

ACADEMIC EDITOR: The reviewers recommend reconsideration the manuscript with revision and modification. I invite the authors to resubmit the manuscript after addressing the comments raised by the reviewers.

We look forward to receiving your revised manuscript.

Kind regards,

Dhanamjayulu C, Ph.D & Post.Doc

Academic Editor

PLOS ONE

Responses: We are extremely grateful to you for giving us the opportunity to revise our manuscript when the final recommendations of the two reviewers are far apart. We have carefully read the comments and questions of both reviewers. According to the 2nd reviewer's two major comments, we have made the relevant supplementary analysis of our data. We used the “Track Changes” function to mark up our revisions. Please see the revised manuscript.

REVIEWER REPORTS

Reviewer Comments:

Reviewer 1

The authors present a paper on the design of interconnects for planar Solid Oxide Fuel Cells geometry. Such topic is very important for the SOFC technology since the Interconnect is a critical component that has to ensure optimal gas distribution and electrical connection in order to prevent additional losses in efficiency of the system.

This work is providing 2D simulation of several parameters as the gas velocity, pressure, and hydrogen molar fraction distribution.

Responses: Many thanks to the reviewer for the above comments. According to the comments, we give the explanations and we have corrected the inaccurate information in the revised manuscript.

1. Authors compare the calculations performed on a “conventional” interconnect and a “novel” interconnect design, however the “conventional” design is just a simple geometry that is actually not in use for real systems. The “novel” design is something closer to the state-of-art interconnect design being characterized by the presence of channels with the function of distributing gases and contacts point aiming at collecting electrical currents. Even neglecting the feasibility of the proposed “novel” design, characterized by sharp edges and abrupt variations in geometry, which is not compatible with common manufacturing methods, I would expect a study on the electrical contribution too. The effect of geometry on the current collection should not be neglected. In fact, authors state this in the Section 1.3 but such aspect is not provided.

Responses: Thanks to the opinions of the referees. Based on several suggestions made by the reviewers, we have modified them.

2. Figure 3.1 is the same as Figure 2.3, I’m not sure is necessary to provide the same figure two times. It would be rather appropriate, for the sake of clarity, to add text boxes into the figures indicating the position and details of the sketch (for example:, rib, channel, electrode…). Also, the flow direction should be indicated by an arrow in the sketches, or at least an indication of the inlet and outlet positions.

Responses: The reviewer's suggestions are greatly appreciated. According to the reviewer's comments, we have made a correction to the picture. 

3. In figures 4.1, 4.2, 4.3, to highlight the differences in the designs it would be helpful to indicate the quotes that are changing. It would be also helpful to indicate where the interconnect is in contact with the electrode and with the gas flow.

Responses: Thanks to the reviewer's suggestion, We have made modifications according to the reviewer's comments.

4. Several graphs lack the axis labels.

Responses: We have made modifications according to the reviewer's comments. 

Reviewer 2

The findings and results in this study are meaningful.

The article provides an idea for a novel 2D simulation model for interconnector SOFCs, aiming to enhance

their performance. The findings and results in this study are meaningful. The introduction, results and conclusions of the manuscript are clear and well-written from the reviewer's perspective.

Although the authors needs point out more the differences between this article and previous research in the introduction, also the authors should reduce the similarity rate of the paper which is 37% in the current version.

Response: We are extremely grateful to the reviewer for the positive comments.

---

## [Editor Report · Decision Letter 1]

23 Jan 2024

Optimized design of planar solid oxide fuel cell interconnectors

PONE-D-23-31430R1

Dear Dr,

We’re pleased to inform you that your manuscript has been judged scientifically suitable for publication and will be formally accepted for publication once it meets all outstanding technical requirements.

Kind regards,

Dhanamjayulu C, Ph.D & Post.Doc

Academic Editor

PLOS ONE

Additional Editor Comments (optional):

The authors are addressed comments properly and it can be accepted for the publication in current form
---

## [Editor Report · Acceptance letter]

30 Apr 2024

PONE-D-23-31430R1 

PLOS ONE

Dear Dr. Shao, 

I'm pleased to inform you that your manuscript has been deemed suitable for publication in PLOS ONE. Congratulations! Your manuscript is now being handed over to our production team.

Kind regards, 

on behalf of

Dr. Dhanamjayulu C 

Academic Editor

PLOS ONE